# Entropy of a bacterial stress response is a generalizable predictor for fitness and antibiotic sensitivity

Zeyu Zhu [1,4], Defne Surujon [1,4], Juan C. Ortiz-Marquez[1], Wenwen Huo [2], Ralph R. Isberg [2], José Bento[3] & Tim van Opijnen [1✉]

Current approaches explore bacterial genes that change transcriptionally upon stress exposure as diagnostics to predict antibiotic sensitivity. However, transcriptional changes are often specific to a species or antibiotic, limiting implementation to known settings only. While a generalizable approach, predicting bacterial fitness independent of strain, species or type of stress, would eliminate such limitations, it is unclear whether a stress-response can be universally captured. By generating a multi-stress and species RNA-Seq and experimental evolution dataset, we highlight the strengths and limitations of existing gene-panel based methods. Subsequently, we build a generalizable method around the observation that global transcriptional disorder seems to be a common, low-fitness, stress response. We quantify this disorder using entropy, which is a specific measure of randomness, and find that in low fitness cases increasing entropy and transcriptional disorder results from a loss of regulatory gene-dependencies. Using entropy as a single feature, we show that fitness and quantitative antibiotic sensitivity predictions can be made that generalize well beyond training data. Furthermore, we validate entropy-based predictions in 7 species under antibiotic and non-antibiotic conditions. By demonstrating the feasibility of universal predictions of bacterial fitness, this work establishes the fundamentals for potentially new approaches in infectious disease diagnostics.

---

[1] Biology Department, Boston College, Chestnut Hill, MA 02467, USA. [2] Department of Molecular Biology and Microbiology, Tufts University School of Medicine, Boston, MA 02111, USA. [3] Computer Science Department, Boston College, Chestnut Hill, MA 02467, USA. [4] These authors contributed equally: Zeyu Zhu, Defne Surujon. ✉email: vanopijn@bc.edu

It is generally assumed that in order to overcome a stress, bacteria activate a response such as the stringent response under nutrient deprivation[1–3] or the SOS response in the presence of DNA damage[4,5]. Measuring the activation of a specific response, or genes associated with this response, can thereby function as an indicator of what type of stress is occurring in a bacterium. For instance, *lexA*, encoding a master regulator of the SOS response in *Escherichia coli* and *Salmonella*[6,7], is upregulated in response to fluoroquinolones, indicative of the DNA damage resulting from this class of antibiotics[7]. Moreover, genes implicated in a stress response can help construct statistical models for predicting growth/fitness outcomes under that stress. For instance, gene-panels have been assembled from transcriptomic data to predict whether a bacterium can successfully grow in the presence of specific antibiotics[8–12]. This type of prediction of growth under antibiotic conditions can lead to point-of-care diagnostics that guide decisions on antibiotic prescription[13].

While methods that are based on a known stress–response or a gene-panel can be valuable in determining a bacterium's sensitivity to a stress, these methods have limited applicability: they only work for small sets of strains, species or environments. For instance, responses such as the stringent or SOS response are only well characterized in a small number of species, genes in a gene-panel may not be present in other strains or species, and responses are not necessarily regulated in the same manner in different strains or species[14,15]. This means that every time such an approach is applied to a new strain, species or condition, a new gene-panel needs to be assembled and validated, which requires the collection of large amounts of data for model training. In contrast, a universal stress response signature would allow for the development of a predictive model that would work for multiple species and conditions, without relying on collecting new data for different settings. While certain organisms may elicit a "general stress response", i.e., regulatory changes coordinated by the same mechanism in response to different types of stress, this general response has not been defined for many species, and it is still not clear to what extent the downstream transcriptional changes triggered under different stress factors overlap[16]. Until this point, there is no generally agreed upon stress response signature that performs as a fitness predictor, with equal or better performance than the gene-panel approaches.

One possible key ingredient in building a universal predictor is to base a prediction not on specific genes, but rather on a bacterium's global response to stress. A global, genome-wide stress response can be captured on at least two organizational levels; RNA-Seq captures transcriptional changes, while transposon-insertion sequencing (Tn-Seq) characterizes the phenotypic importance of genes, i.e., a gene's contribution to fitness in a specific environment[17–22]. We have previously shown that when an organism is challenged with an evolutionarily familiar stress (i.e., one that has been experienced for many generations), it triggers a subtle response, whereas the response becomes more chaotic when the bacterium responds to a relatively unfamiliar stress, for instance antibiotics[17]. This suggests that the degree to which a bacterium is adapted to a specific stress may be predicted from the global response it elicits. It is possible to observe genome-wide differences between stress-susceptible and stress-resistant bacteria in data from previously published transcriptomic studies that mostly focus on gene-panel approaches. Specifically, in these data it can be observed that the number of differentially expressed genes, and the magnitude of changes in expression seem to be more dramatic in stress-susceptible strains than stress-resistant ones[8–12,23]. Therefore, if these are indeed characteristic differences between responses coming from stress-sensitive and stress-resistant bacteria, and these differences can be appropriately quantified, an opportunity would arise to define a universal method that can predict fitness for multiple species and conditions.

In this study we generate and analyze a substantial transcriptomic dataset for the bacterial pathogen *Streptococcus pneumoniae*. To validate our dataset, existing gene-panel approaches are replicated and scrutinized as a point-of-comparison. Thereby, we first demonstrate that bacterial fitness under antibiotic or nutrient stress can be predicted by expression profiles from small gene-panels, while a separate panel can predict an antibiotic's mechanism of action. We highlight the limitations of these existing approaches by showing that gene-panels are sensitive to model parameters and the data they are trained on, and are limited to strains and species that share the same genes. With the goal to develop a general approach, we explore the observation that global transcriptional disorder seems to be a common stress feature in bacteria. It turns out that increasing disorder stems from an increasing loss of dependencies among genes (e.g., regulatory interactions). These dependencies manifest as correlations in gene expression patterns, and by accounting for these dependencies, the statistical definition of entropy can be used to accurately quantify the amount of disorder in the system. First, we show that when entropy is calculated using time-series RNA-Seq data and dependencies amongst genes are accounted for, stress-sensitive strains have higher entropy than stress-insensitive ones. This enables fitness predictions using a simple decision rule, where if entropy is either above or below a threshold, fitness is respectively low or high. Importantly, this entropy-based method achieves better performance in predicting fitness outcomes compared to existing gene-panel approaches. In order to simplify the approach, we show that entropy can be calculated using a single time-point, and does not necessarily require time-series data to achieve high accuracy. To highlight the universality of entropy, in addition to evaluating performance on a previously unseen test set, validation experiments are performed for seven Gram-negative and -positive pathogenic species, and the approach is applied to multiple published datasets. Moreover, we show that transcriptional entropy is correlated with the level of antibiotic sensitivity, enabling MIC predictions. Overall, we develop a large new experimental dataset, and a species-independent fitness prediction method based on entropy. By carefully defining entropy, we illustrate that entropy does not simply capture large changes in expression, but instead builds upon an intuitive notion of disorder, and enables predictions on bacterial fitness. We present gene-panel based methods as a baseline for comparison, and demonstrate that entropy-based methods perform better, are robust to parameter tuning, and can accommodate different amounts of data to enable fitness predictions. Most importantly, unlike gene-panels, entropy-based predictions generalize to previously unseen settings, and to multiple pathogenic bacteria.

## Results

**Existing methods have several limitations, and do not generalize.** Previously, the expression levels of specific genes have been used to predict susceptibility of a specific species under a specific antibiotic stress[8,11,23]. In contrast, the goal here is to identify a general predictor of fitness (presence or absence of growth) that does not only work for a specific stress or species, but instead extends to as many previously unseen settings (i.e., species and conditions) as possible. We hypothesized that, in line with existing approaches, a gene-panel that predicts fitness could be generated. This panel, when trained on expression data coming from multiple stress conditions, would then predict bacterial fitness for any condition (rather than a specific condition). Importantly, we would thereby also be able to assess how sensitive

such models are to input data and model parameters. Below we first show that gene-panel models indeed are highly sensitive to these factors and thereby have limited generalizability. Subsequently, we develop an alternative approach using entropy, that is generalizable, robust, and condition-agnostic (i.e., applicable to many conditions).

To test the first hypothesis, whether a gene-panel model can be trained that predicts fitness for many different conditions, a large RNA-Seq dataset was generated for the human pathogen *Streptococcus pneumoniae*. To produce transcriptomic response profiles from multiple stress conditions, *S. pneumoniae* strains TIGR4 (T4) and Taiwan-19F (19F) were grown in the presence or absence of 1× the minimum inhibitory concentration (MIC) of 16 antibiotics representing four mechanisms of action (MOA). These include, cell wall synthesis inhibitors (CWSI), DNA synthesis inhibitors (DSI), protein synthesis inhibitors (PSI), and RNA synthesis inhibitors ((RSI); Fig. 1a, Supplementary Tables 1 and 2). Each strain was exposed to each antibiotic for 2–4 h and cells were harvested for RNA-Seq at various time points. As T4 and 19F are susceptible to most antibiotics used, the transcriptional profiles in the presence of antibiotics mostly represent cases of low fitness (Fig. 1a, sensitive strain, 1× $MIC_{WT}$). In order to find patterns that differentiate fitness outcomes, we generated adapted strains with increased fitness in the presence of antibiotics by serial passaging wildtype T4 and 19F in the presence of increasing amounts of antibiotics. Four independent adapted populations for each strain were selected on individual antibiotics. These adapted strains could grow in the presence of antibiotic at 1.5xMIC of the wildtype strain, albeit with a small growth defect (Supplementary Fig. 1a, b). In parallel, RNA-Seq was performed on *S. pneumoniae* strains D39 and T4 in a chemically defined medium, and media from which either uracil, glycine, or L-valine was removed, which are essential for D39 but not T4. This enabled the potential identification of a common stress signature that is shared between antibiotic exposure and nutrient deprivation, and across multiple strains. Lastly, D39 was adapted to grow in the absence of each individual nutrient, after which RNA-Seq was repeated for adapted clones (Supplementary Table 1 lists all 24 strains, 67 populations and 267 RNA-Seq experiments; RNA-seq data is provided in Supplementary Data 1, and it is possible to visualize and explore all data using a ShinyOmics[24] based app online at http://bioinformatics.bc.edu/shiny/ABX).

Transcriptome data were separated into a training set for parameter fitting, and a test set. The test set includes a completely different set of antibiotic conditions, to enable proper evaluation of model performance on previously unseen data (Supplementary Table 1). A condition-agnostic predictor of fitness was developed by fitting a regression model on the training set, which includes high and low fitness outcomes from five antibiotics (representing four MOAs), three nutrient depletion conditions, and from three *S. pneumoniae* strain backgrounds. Lasso-regularization was used in order to limit the number of features, thereby lowering the risk of overfitting the model (there are over 1500 genes in common for the three strains, therefore there are as many potential features that could be used)[25]. In order to avoid any bias in the selection of features, the regularization strength ($\lambda$) was automatically determined using crossvalidation analysis on the training data (Fig. 1b)[25,26]. The resulting model (which contains 28 genes and an intercept, Supplementary Table 3) has an accuracy of 0.93 and 0.77 on the training and the unseen test set, respectively (Fig. 1c, Supplementary Fig. 2, full performance statistics are in Supplementary Data 6).

Fitness predictions that rely on the expression of specific genes are potentially influenced by the data used during training[23]. A model robust to input data would recover mostly the same features (i.e., genes) when small subsets of input are omitted

during parameter fitting. In order to test the sensitivity of the regression model to input data, the same type of regression model was trained on five different subsets of the training dataset, each time omitting a different 20% of the data. The features included and their coefficients varied greatly in these experiments (Fig. 1d), with only 5 out of 28 genes in the model common to all iterations of model fitting. To assess sensitivity of the gene-panel to the regularization strength (i.e., $\lambda$), the same model was trained using different values for $\lambda$. While the coefficients of individual genes vary drastically (Fig. 1e), the performance at different values of $\lambda$ remains similar (Fig. 1b, Supplementary Fig. 2B). This indicates that there are genes that contain similar information for classification purposes, and are interchangeable. Thus, we demonstrate that the gene-panel approach is sensitive not only to input data, but also to model parameters. An implication of this sensitivity is that the genes in a gene-panel that are selected in an automatic fashion can be influenced by how the model is trained. Therefore, interpreting these genes as the determinant biological factors for fitness can be problematic. Furthermore, enrichment analysis reveals there are no significantly enriched functional categories in this gene-panel (Supplementary Fig. 2E). This suggests that a gene-panel is not a suitable approach for developing a condition-agnostic model, since no specific common response to different stresses can be detected that separates low fitness cases from high fitness ones.

While a condition-agnostic gene-panel is sensitive to input data and model parameter $\lambda$, it remains to be seen whether condition-specific models suffer from the same issue as well. For three MOA's for which we generated data for multiple antibiotics (CWSI, DSI, and PSI), regularized regression models were trained (Supplementary Table 4), and the models' sensitivities to input data and $\lambda$ were evaluated. In all three cases, the models change with input and $\lambda$, and show no enrichment for specific functional categories (Supplementary Fig. 3). In contrast, some published gene-panels[11] have shown functional enrichment (Supplementary Data 2). However, this is likely because the published gene-panels have been developed for single antibiotics. Therefore, the genes in those panels are highly selective for the species-specific response that is triggered in a particular stress. In contrast, in this work, we identify predictors that differentiate high and low fitness cases for multiple stresses. The fact that there is no enrichment on our gene-panels is suggestive of a lack of a general response, characterized by a set of specific genes, that gets triggered under many different circumstances.

Besides a lack of functional enrichment, neither the MOA-specific nor the condition-agnostic gene-panels developed here include genes that are known direct-targets of the antibiotics used. Moreover, in addition to being sensitive to input data and regularization strength, the condition-agnostic fitness gene-panel is limited in its applicability to other species, as genes in this panel lack homologs in other Gram-positive as well as Gram-negative species (Fig. 1f, Supplementary Table 5). In fact, this homology problem is a limitation of previously published gene-panels as well (Fig. 1g, Supplementary Table 5). Gene-panel based models therefore not only require re-training for each new condition, but also when they are to be implemented for a new species. This shows that gene-panel approaches in general not only need to be applied and interpreted with caution, but there is also no good evidence to expect that they can be turned into a generalizable fitness predictor that is both species and condition-agnostic.

**Gene-level transcriptional responses are unique to the type of stress.** We hypothesized that one of the reasons why it may be nontrivial to produce a condition-agnostic model is because the different conditions (i.e., MOA's of different antibiotics) trigger

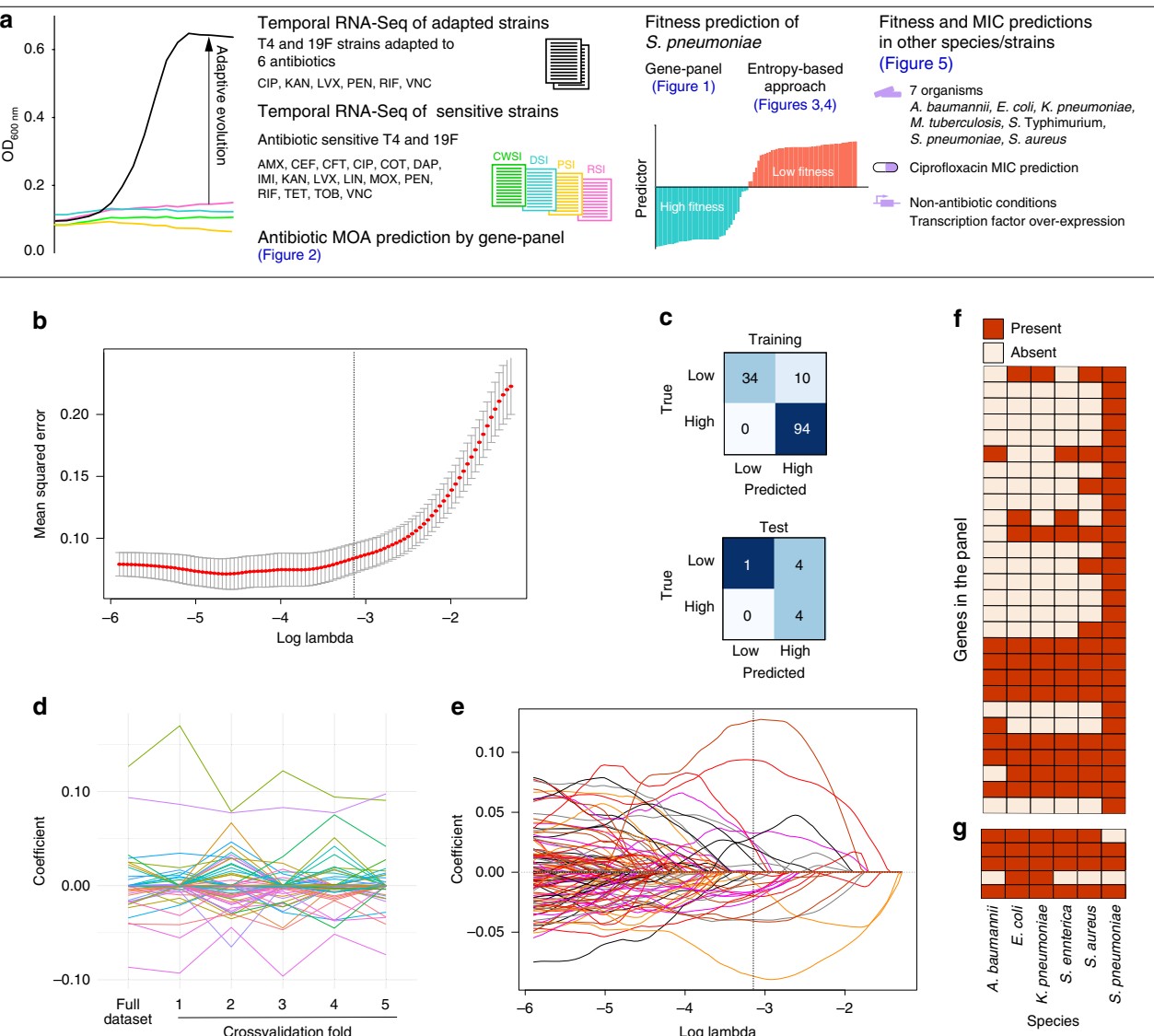

**Fig. 1 Gene panel-based fitness predictions of *S. pneumoniae* under antibiotic and nutrient stress. a** Project setup and overview. Wildtype and adapted strains of *S. pneumoniae* are exposed to multiple antibiotics, belonging to four different classes, and their fitness outcomes in each condition is determined by growth curves. Temporal RNA-Seq data is used to train models that predict the MOA of an antibiotic, and the fitness outcome of a strain using gene-panel approaches. The concept of entropy is developed expanding predictions to MIC and fitness for other strains and species in the presence of antibiotics and in non-antibiotic conditions. CWSI cell wall synthesis inhibitors: AMX amoxicillin, CEF cefepime, CFT ceftriaxone, IMI imipenem, PEN penicillin, VNC vancomycin; DSI DNA synthesis inhibitors: CIP ciprofloxacin, COT cotrimoxazole, LVX levofloxacin, MOX moxifloxacin; RSI RNA synthesis inhibitor: RIF rifampicin; PSI protein synthesis inhibitors: KAN kanamycin, LIN linezolid, TET tetracycline, TOB tobramycin; DAP daptomycin (a membrane disruptor). **b** A gene-panel for fitness prediction is generated by a regularized logistic regression model fit on differential expression data from the training set. The selected value of $\lambda = 0.0428$ is shown as a dashed line, resulting in 28 genes in this panel. Red points and error bars represent mean ± standard deviation of error across $n = 5$ crossvalidation folds. **c** Prediction performance of the fitness gene-panel is shown as confusion matrices for the training (top) and test (bottom) datasets. The gene-panel generates 10 and 4 false positives, and an overall accuracy of 0.93 and 0.77 in the training and test data sets respectively. **d** Coefficients of individual features (i.e., genes) are plotted for the model trained on the full dataset, and five crossvalidation training folds, where 20% of the data is omitted during model fitting. The gene-panel is highly affected by training data, indicated by many genes having nonzero coefficients on some folds, but not others. Only 5 out of the 28 genes in the fitness gene-panel are maintained as predictors in the regression models across all folds. **e** Each gene's coefficient is plotted as an individual line, against varying values of $\lambda$. The gene panel is highly affected by $\lambda$, indicated by the nonmonotonic increase or decrease in the coefficient in each gene. In fact, there are many genes that have nonzero coefficients only for a small range of $\lambda$. Dashed line depicts the selected value of $\lambda$ as in (**b**). **f** The presence and absence of each of the 28 genes in the *S. pneumoniae* fitness panel is highly variable across 5 Gram-positive and Gram-negative species. **g** A published *E. coli* ciprofloxacin sensitivity panel[11] also suffers from a lack of conservation across the same group of species. Gene identifiers can be found in Supplementary Table 5.

such distinct responses that it is unlikely to identify a common signature among them. To determine whether responses from different antibiotics that fall under the same MOA cluster together, principal component analysis (PCA) was performed on the complete differential expression dataset. Each experiment is presented as one trajectory, connecting individual timepoints within that experiment (Fig. 2a). *K*-means clustering of all experiments' trajectories showed that transcriptional responses to drugs within the same MOA tend to follow similar trajectories over time (Fig. 2a, b).

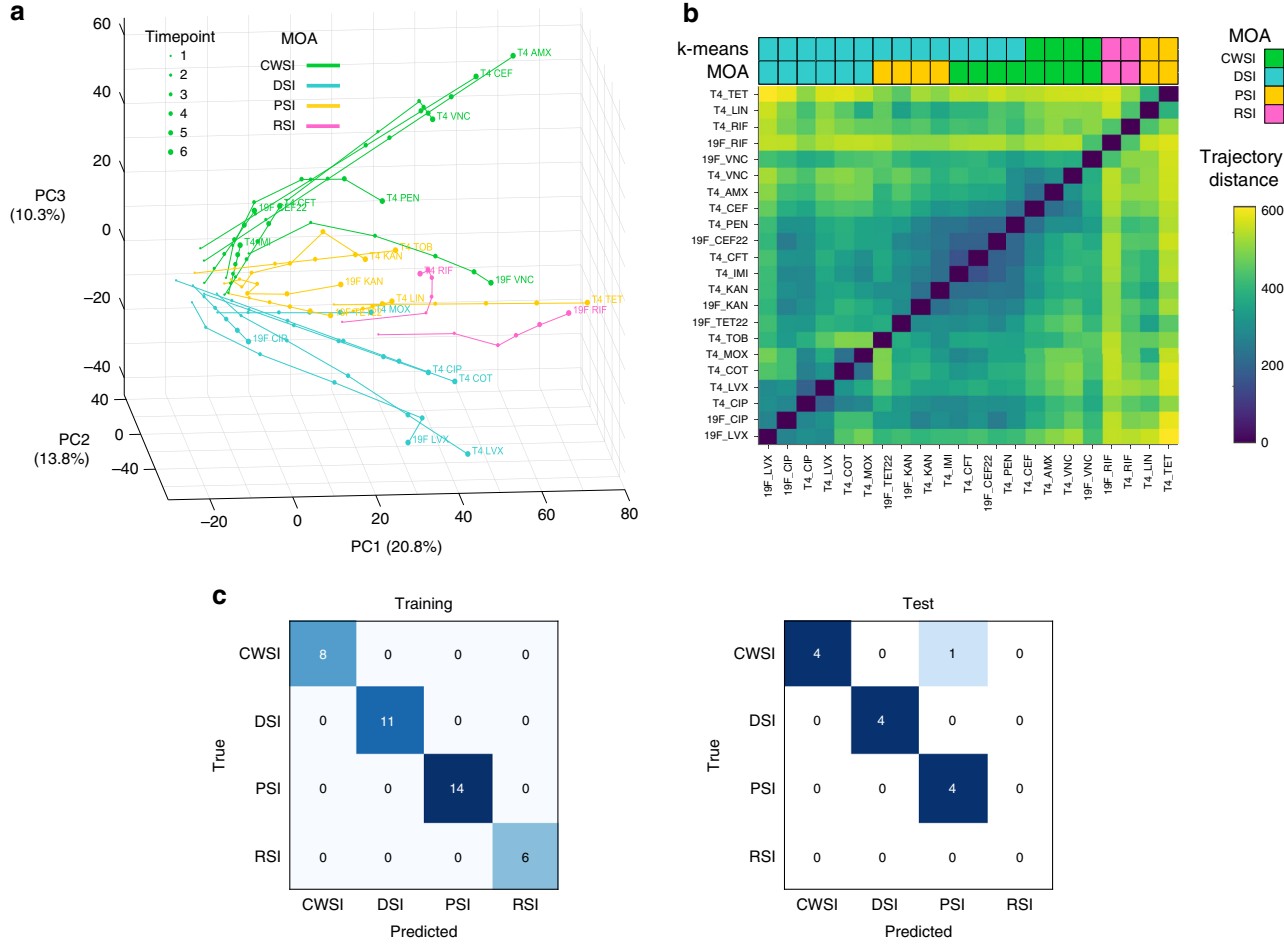

**Fig. 2 Transcriptional responses separate antibiotics with different mechanisms of action. a** Principal component analysis (PCA) on differential expression datasets from sensitive *S. pneumoniae* strains T4 and 19F grown in the presence of 16 different antibiotics at 1× MIC depicts antibiotic responses as temporal transcriptional trajectories. Each line describes the trajectory of one of one strain in the presence of a CWSI (AMX, CEF, CFT, IMI, PEN, VNC), DSI (CIP, COT, LVX, MOX) PSI (KAN, LIN, TET, TOB), or RSI (RIF). Trajectories for each strain are largely grouped based on their MOA, and grouped-trajectories become more distinct over time. The size of each data point increases with the time of antibiotic exposure; each trajectory is split into 6 timepoints, e.g., for an experiment that spans 120′ each point indicates a 20′ increment. Abbreviations are as in Fig. 1. **b** In order to quantify the separation of the PCA trajectories by an antibiotic's MOA, pairwise distances between PCA trajectories were computed (see Methods). Pairs of transcriptional trajectories obtained using drugs within the same MOA tend to have smaller distances than pairs obtained using drugs with different MOA's. *K*-means clustering of the trajectory distances groups the trajectories mostly by MOA, although some PSI and CWSI trajectories are grouped with DSI ones. The top and bottom bars above the heatmap show the *K*-means clustering result, and the real MOA of each trajectory respectively, which have 64% agreement. **c** Confusion matrices indicating the performance of the gene-panel that predicts MOA. This panel was generated using a multi-class regression model (see Supplementary Table 1 for the training and test set split, and Methods for details on parameter tuning) and consists of 34 genes. The gene-panel correctly predicts the MOA on all training set data and only misclassifies a single experiment on the previously unseen test dataset, showing the different MOA's being easily distinguishable with simple gene-based methods.

To further analyze whether different MOA's trigger different responses, a multi-class logistic regression model was fit on the training dataset, and evaluated on the test set. If a simple classifier can successfully distinguish between different MOA's, this would imply that there are discriminating signals specific to each MOA. Similar to the fitness prediction, the regularization parameter was selected via a principled automatic procedure (without making any arbitrary decisions) to avoid overfitting (Supplementary Fig. 4A). This simple regression model is able to classify MOA's with an accuracy of 1 on the training set, and with only a single misclassification in the test set (Fig. 2c, Supplementary Fig. 4D). Similar to our fitness panel, enrichment analysis of the 34 genes in this MOA panel reveals no significantly enriched functional categories (Supplementary Fig. 4E). While some of the genes in the panel are relevant to the action of specific antibiotics, it is not

immediately evident how each individual gene is relevant for the classification. For instance, DNA gyrase A (SP_1219) appears in the MOA panel (Supplementary Data 4), and is a direct target of fluoroquinolones LVX and CIP, belonging to the class DSI. However, it is downregulated to a higher extent under both RSI compared to DSI stress, and thus does not have much discriminating power on its own (Supplementary Fig. 4D). Compared to the fitness prediction panel, the features in the MOA panel are more robust to parameter tuning (Supplementary Fig. 4B), and to input data (Supplementary Fig. 4C). This suggests that MOA prediction is an easier task than fitness prediction using existing gene-panel approaches. Previous studies have demonstrated it is possible to train a classifier that predicts MOA from whole transcriptome data[27,28]. However, it was unclear whether MOA could be predicted from the expression of a few

genes. Our model could therefore, for instance, be implemented to classify the MOA of novel antimicrobials, without having to profile the entire transcriptome.

**Entropy as a measure of transcriptional disorder predicts fitness.** While the practical application of the MOA model may be useful, the main goal of this work is to build a versatile toolbox for fitness predictions that does not have many parameters to tune, does not rely on specific genes, and therefore possibly has improved generalizability compared to gene-panel models. To accomplish this, we focused on the following observation that we made in the data presented in this work, as well as in previously published studies[11,12,23,29]: bacteria with low-fitness in a given condition trigger larger, and seemingly more chaotic gene expression changes than those with high fitness (Fig. 3a, b). Specifically, the temporal response of the wildtype strain with low fitness shows an escalating response over time, with increasing and fluctuating transcriptional changes. In contrast the response of the adapted strain, with high fitness, is contained with only small changes in expression (Fig. 3a). Since these characteristics can be observed for many different stress-types and species, it could possibly be turned into a generalizable predictor of fitness if appropriately captured. Importantly, these types of patterns in the data evoke statistical entropy, which is a well-established concept that captures the amount of disorder in a system (Fig. 3b, Supplementary Fig. 5). Figure 3b shows three hypothetical scenarios. Genes in scenarios 1 and 2 have some sort of regulatory interaction, for instance because they are in the same operon. In scenario 2, the individual genes' expression patterns have differences in magnitude and direction, but all genes still have similar overall expression trajectories that co-vary. Therefore, the first 2 scenarios are illustrative of strong dependencies among genes. In contrast, scenario 3 highlights a more disordered pattern, and a lack of dependencies between genes, which results in this scenario's entropy being the highest. We hypothesized that with increasing amounts of stress (i.e., when the fitness of the bacterium is lowered), the bacterium experiences increasing amounts of dysregulation, resulting in a loss of dependencies in expression among genes. A loss of such dependencies results in more and more genes changing in expression independently (and perhaps seemingly randomly), resulting in an increase in entropy. Based on this idea, we aimed to quantify the amount of disorder in a transcriptomic response by computing entropy. To predict fitness, we then use a simple decision rule on a single feature, which avoids overfitting, where entropy higher than a threshold $t$ predicts low fitness, and entropy lower than $t$ predicts high fitness.

To calculate entropy on a transcriptomic dataset with multiple timepoints, we redefine the classical statistical concept of entropy ($H$) as follows:

$$H = \ln(|\Sigma_\rho|), \quad (1)$$

Where $\Sigma$ is the empirical covariance matrix ($\Sigma_p$ is the empirical covariance of gene$_i$ and gene$_j$ computed from the time series data), and $|\Sigma|$ denotes the determinant of $\Sigma$[30–33]. $\Sigma_p$ is a graphical-lasso regularized $\Sigma$, where $\rho$ denotes the regularization strength.

Entropy is computed from experiments with multiple timepoints as follows. (1) The temporal differential expression (DE) data is used to compute a gene–gene empirical covariance matrix $\Sigma$. (2) Graphical lasso[34] is applied to $\Sigma$ to obtain a regularized inverse of this covariance matrix ($\Sigma_p{}^{-1}$). The matrix $\Sigma_p{}^{-1}$ represents a network of dependencies of the regulatory interactions of the genes. (3) The inverse of this matrix ($\Sigma_p$) can then be used in Eq. (1) to compute entropy (Supplementary Fig. 5).

It is important to note that, with the described approach, a high entropy response reflects large changes in magnitude in the

transcriptome that come from independently responding genes. This means that large changes in magnitude can still result in low entropy, when changes in expression are synchronized among genes (Fig. 3b). Synchronization thus comes from dependencies between genes, for instance due to regulatory interactions, which can vary based on the condition. Here, it is assumed that there is a sparse network of such dependencies (i.e., regulatory interactions), which are specifically determined for each experimental condition. These regulatory interactions for each experiment are inferred by computing a covariance matrix $\Sigma$ from temporal DE data. The inverse of this covariance matrix ($\Sigma^{-1}$) is interpretable as the (condition-specific) regulatory interaction network, where gene pairs have a zero value on $\Sigma^{-1}$ when their expression patterns are not directly dependent on each other. Like most biological networks, the condition-specific regulatory interaction network is expected to be sparse[35–37]. However, raw values on $\Sigma^{-1}$ empirically measured using RNA-Seq data, are mostly nonzero, resulting in a dense network, potentially due to noise in data collection. Regularization is thereby applied on $\Sigma^{-1}$ to estimate a de-noised, sparse network of interactions $\Sigma_p$, more likely to represent real, biologically relevant regulatory dependencies.

Training of this multi time-point entropy model includes the determination of two parameters: regularization strength $\rho$ and threshold $t$. This is accomplished by first determining $\rho$ by fivefold crossvalidation (on the training set), and then determining $t$ for this selected $\rho$. $\rho$ at 1.5 minimizes crossvalidation error (Fig. 3c), and using this value of $\rho$ on the full training set, results in a threshold $t$ of 1066.25. This in turn yields an accuracy of 0.97 and 0.84 in the training and test sets respectively (Fig. 3f, Supplementary Datas 5 and 6), which are both higher accuracies than the corresponding values obtained with the gene-panels (Fig. 1, Supplementary Data 6). Receiver–operator characteristic (ROC) curve analysis shows that entropy can effectively separate high and low fitness cases, with an area under the ROC curve of 0.99 and 0.91 for the training and test sets, respectively (Fig. 3d). Precision-recall (PR) curve analysis reveals that entropy can detect high-fitness cases, with an area under the PR curve of 0.99 and 0.98 for the training and test sets, respectively (Fig. 3e). Both ROC and PR analyses thus show much better performance of entropy compared to the gene-panel on the test set (Supplementary Data 6). Moreover, entropy of each cellular function is similar for a given experiment (Supplementary Fig. 6), suggesting that transcriptome-wide entropy is not dominated or influenced by a certain set of genes. Unlike the gene-panel based fitness prediction models, the entropy model is robust to the selection of regularization strength $\rho$. It is possible to set $\rho$ to be an extreme value and still get comparable performance to the model above (Supplementary Fig. 7). Here, two such extreme values are considered. For instance, if $\rho = \infty$ (i.e., the co-variances among genes are ignored and genes' responses are assumed to be independent), entropy can be computed as the average of the logarithm of variances of all genes (see "Methods"). In this case, the training and test set accuracies are 0.94 and 0.74, respectively (Supplementary Data 6), which is comparable to the fitness gene-panel. If, on the other extreme, $\rho = 0$, i.e., entropy is computed directly on the non-regularized covariance matrix, the model will over-correct for a dense network. In this case, the training and test set accuracies are 0.86 and 0.32, respectively (Supplementary Data 6). In this case, the poor performance on the test set is likely due entropy being sensitive to the number of experimental timepoints used. The training set (which is used for determining $t$) contains mostly experiments with seven timepoints or more, whereas the test set contains experiments with only two timepoints (Supplementary Table 1). Using fewer timepoints changes entropy in the same direction for most experiments,

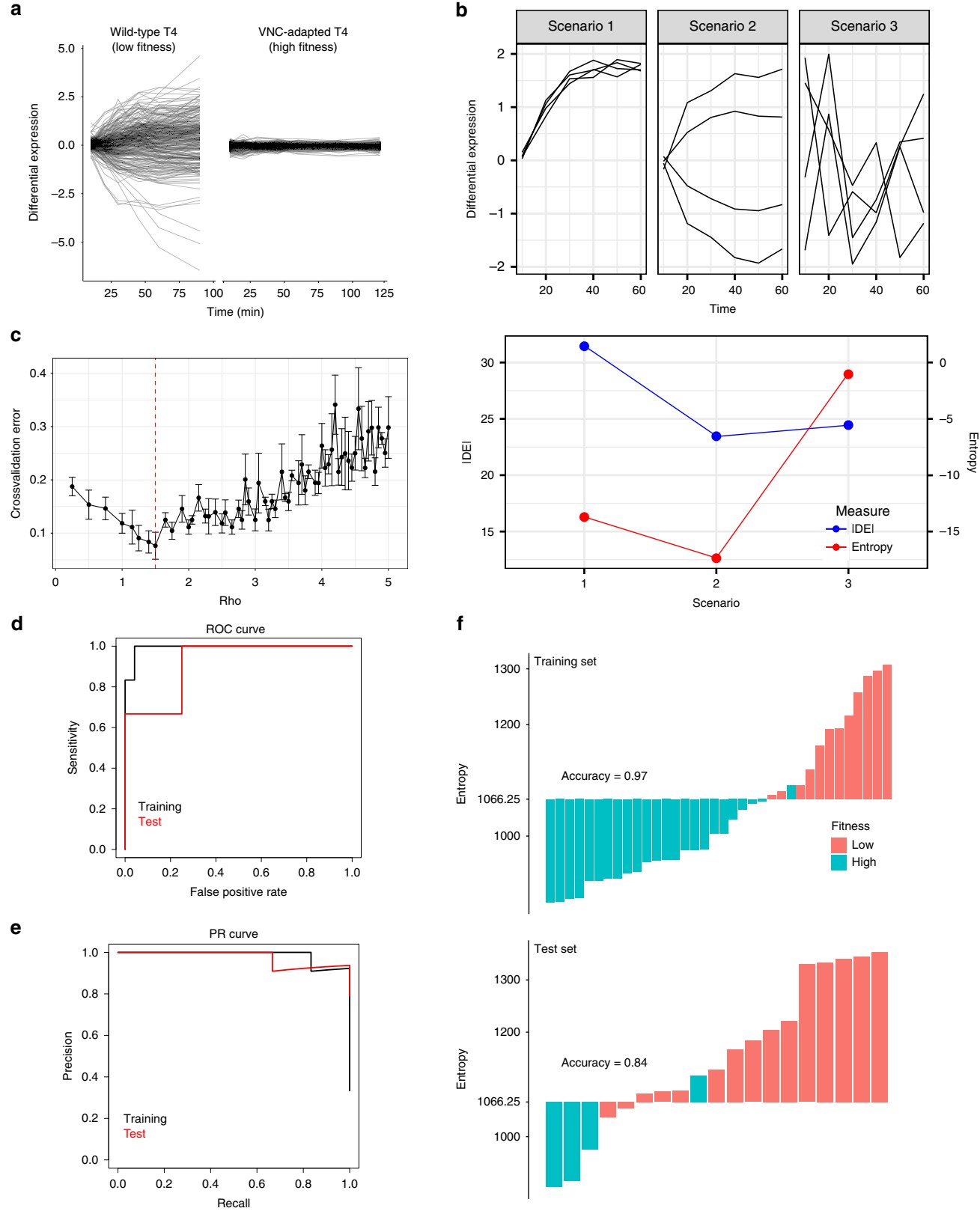

while maintaining a separation between high and low-fitness cases: for $\rho = 0$ or $\rho = \infty$ entropy in most experiments is increased when more timepoints are used, whereas for $\rho = 1.5$ entropy in most experiments is decreased when more timepoints are used (Supplementary Fig. 8). For $\rho = \infty$ or $\rho = 1.5$ this effect does not negatively impact predictive performance drastically (Fig. 3,

Supplementary Fig. 7). However, for $\rho = 0$ it appears that the value of $t$ determined on the training set is inappropriate for the test set. Yet the low-fitness experiments in the test set still have higher entropy than high-fitness experiments (Supplementary Fig. 7C). Thus, a lower threshold for entropy could perform better on experiments with fewer timepoints. While the model is

**Fig. 3 Transcriptomic disorder can be quantified by entropy, which predicts fitness. a** Depiction of the transcriptomic response of wildtype T4 and VNC-adapted T4 in response to 1× MIC-wt of Vancomycin. Differential expression (DE) of each gene over time is represented as a line. The response of the wild type is more disordered than the adapted-response, and has higher entropy. **b** Entropy captures disorder in a transcriptome and not simply high-magnitude changes. The top panel shows three hypothetical scenarios, where DE of four individual genes are tracked over time. In scenarios 1 and 2, the individual genes are dependent on each other and follow similar transcriptional trajectories. In scenario 3, dependencies are largely absent and the overall changes in DE seem much more disordered. In the bottom panel, magnitude changes (blue, quantified as the sum of absolute DE), and entropy (red) for the three scenarios are compared. While the largest changes in magnitude are in scenario 1, both scenario 1 and 2 have relatively low entropy, due to dependencies among genes. In scenario 3, overall DE is similar to the other two scenarios, but the magnitude changes have lost much of their dependency and have become disordered, resulting in high entropy. **c** Selection of regularization parameter $\rho$. Fivefold crossvalidation was used to determine the best choice of $\rho$. Error (1-accuracy) is reported as the mean ± standard deviation across $n = 5$ folds. The value of $\rho$ that minimizes the mean crossvalidation error is determined to be 1.5 (red dashed line). **d** Performance of temporal entropy-based fitness prediction is shown as receiver-operator characteristic (ROC) curves plotting the sensitivity against the false-positive rate across a range of thresholds for training (black) and test (red) datasets. The area under the ROC (AUROC) curve shows how well the predictor can separate high and low fitness. The AUROC is 0.89 and 0.94 for the training and test set respectively. **e** Performance of temporal entropy-based fitness prediction is shown as precision-recall (PR) curves plotting precision against recall across a range of thresholds for training (black) and test (red) datasets. The area under the PR curve (AUPRC) shows how well the predictor can detect high fitness cases. The AUPRC is 0.88 and 0.98 for the training and test set respectively. **f** Entropy of all experiments in the training (top panel) and test (bottom panel) sets. Each experiment is represented as an individual bar, colored according to the experimentally determined fitness outcome. Bars above the entropy threshold (Entropy = 1066.25) are predicted to be low fitness and bars below the threshold are predicted to be high fitness. Both training and test sets score very well with an accuracy of 0.97 and 0.84, respectively.

sensitive to extreme changes in regularization, this sensitivity is not as severe as the gene-panels, since the extreme value of $\rho = \infty$ also yields a test set accuracy of 0.74, which is comparable to the gene-panel method with a 0.79 test set accuracy. The entropy-based model thus operates with highest accuracy when biologically realistic assumptions are made.

**A simpler model of entropy predicts fitness from a single timepoint.** The time course experiments accurately capture a bacterium's survival in a test environment, but they are labor intensive and potentially expensive. In cases where temporal information may not be available or is prohibitively expensive to generate, computing covariance across genes is not possible. However, entropy can still be determined for a single-timepoint transcriptome profile as follows[38]:

$$H_{stp} = \ln(\sigma^2), \qquad (2)$$

where $\sigma^2$ is the variance of the distribution of DE across genes for a single timepoint (Fig. 4a, b). This simpler definition of entropy enables the approach to be applied even in settings where temporal transcriptional information cannot be obtained. Similar to the temporal models, a threshold for entropy was determined automatically (in this case 2.08), which is the value that maximizes classification accuracy in the training set which contains data from multiple timepoints. Analogous to the temporal models, low fitness is associated with higher entropy compared to high fitness conditions (Fig. 4, Supplementary Data 3). The single-timepoint variant of entropy outperforms gene-panels: on the test set, the area under ROC curve is 0.88 for entropy, and 0.75 for the gene-panel (Fig. 4d, Supplementary Data 6). Similarly, for the test set, the area under the PR curve is 0.96 for entropy, whereas for the gene-panel, it is 0.32 (Fig. 4e, Supplementary Data 6). Moreover, the single timepoint variant of entropy can classify low and high fitness cases with an accuracy of 0.81 and 0.61 in the training and previously unseen test sets respectively (Fig. 4f, Supplementary Data 6). However, our data shows that different antibiotics trigger responses in a time dependent manner, which may lead to ambiguities in the entropy-based prediction of fitness for early timepoints for antibiotics that cause a slower response (e.g., KAN, Fig. 4c). Therefore, predictions based on (slightly) later timepoints might result in improved accuracy. To test this, the training and test datasets were split into early (≤45 min of stress exposure) and late (≥60 min of exposure) timepoints. Two new thresholds for entropy

were determined: $t_{early} = 0.94$ on the early timepoints and $t_{late} = 2.11$ on the late timepoints within the training data. On the early timepoints, $t_{early}$ achieves an accuracy of 0.75 and 0.63 on the training and test sets, respectively. On the later timepoints, $t_{late}$ yields a high accuracy of 0.88 and 0.84 on the training and test datasets, only including three false-positive predictions in the test data set (Fig. 4g). This shows that entropy computed on data from later time points results in a higher predictive accuracy of fitness outcome than earlier time points (Fig. 4g). Biologically this also makes sense, because while only some antibiotics trigger a clear response within 30–60 min after exposure, all antibiotics trigger an increasingly pronounced response as exposure times progress past 60 min. The time dependency of an antibiotic response thus makes it more difficult to accurately predict fitness using data from early timepoints. This time dependency would affect the gene-panel for fitness predictions as well. Even though the gene-panel is trained and tested on only the later timepoints and has far poorer performance compared to entropy trained and tested on the same (late) timepoints. Moreover, entropy trained on early timepoints does only slightly worse than gene-panels trained on late timepoints, with only three additional misclassifications (Supplementary Data 6). This highlights that despite the time dependency of an antibiotic response, our new entropy-based approach can make predictions on at least two time frames, unlike gene-panels.

Overall, the entropy model (and its variants) has several advantages. First, it is based on a simple, and intuitive principle: large and independent changes in the transcriptome are indicative of dysregulation, and beyond a threshold predictive of low fitness. Second, it is possible to simplify the entropy-based model to accommodate less data (i.e., single timepoint transcriptome). Third, an entropy-based model has few parameters (at most two parameters need to be determined), and is therefore less likely to be overfit to data. Fourth, the model does not depend on the identity of specific genes, who may or may not be present in different strains/species. Fifth, the model could be easily applied to other data types (e.g., proteomics and metabolomics). Therefore, an entropy-based model is more likely than a gene-panel based approach to be generalizable to previously unseen conditions and species.

**Entropy-based predictions generalize across species and conditions.** To test if the entropy-based approach is indeed generalizable and successfully predicts fitness for other *S.*

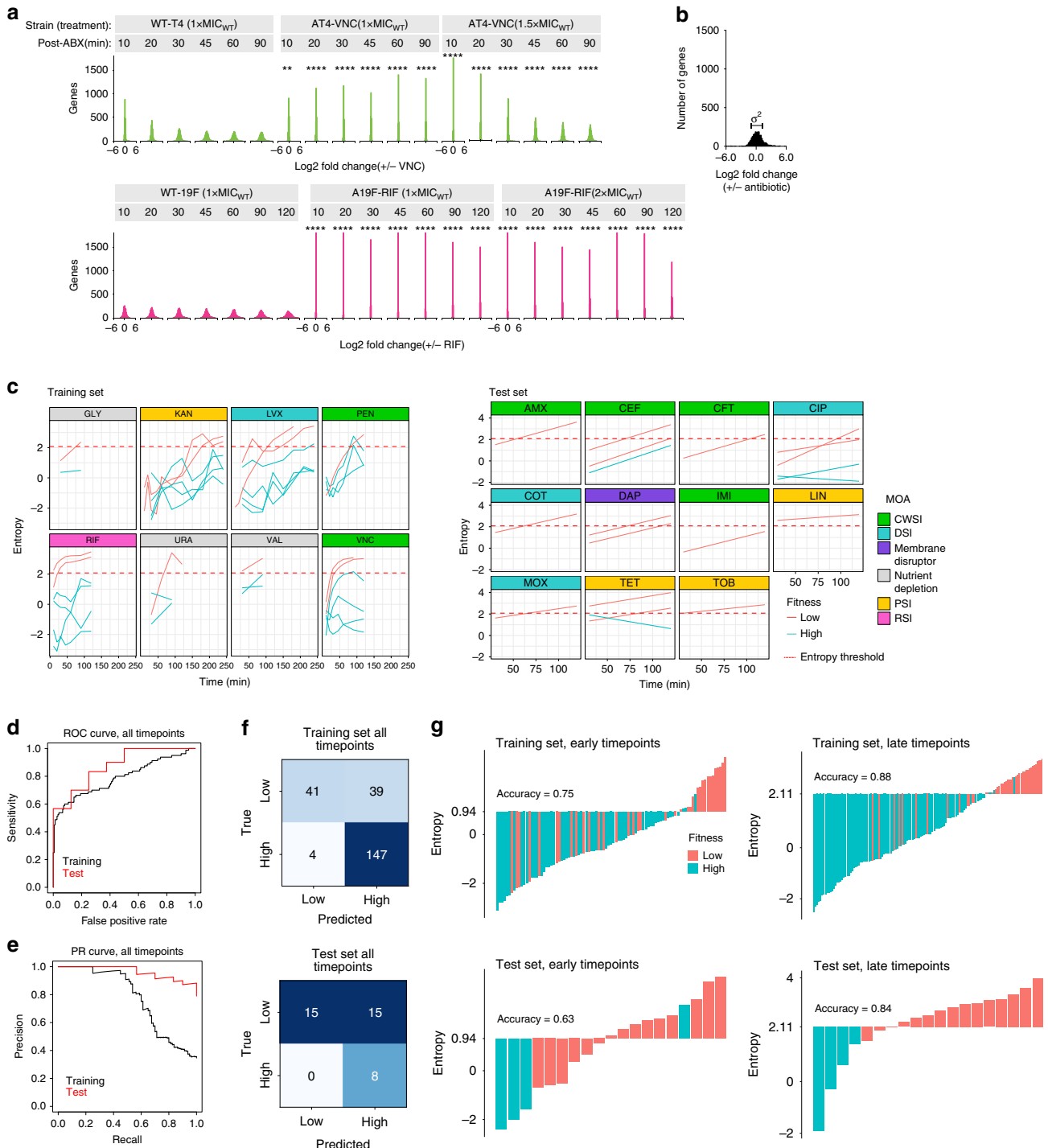

pneumoniae strains and other species, a new RNA-Seq dataset was generated under ciprofloxacin exposure for *Salmonella typhimurium*, *Staphylococcus aureus*, *E. coli*, *Klebsiella pneumoniae*, and two additional *S. pneumoniae* strains representing serotypes 1 and 23F (Supplementary Table 1). These five species represent both Gram-negative and Gram-positive bacteria and cover a wide range of ciprofloxacin MICs (Fig. 5a). Since the single-timepoint variant of entropy is the most practical (in terms of data collection and cost), the generalizability of entropy to previously unseen species was evaluated using this model. RNA-Seq was performed at 120 min post exposure to 1 µg per mL of CIP. The overall response characteristics are similar to what was observed for *S. pneumoniae*, with 120 min exposure to 1 µg per

mL ciprofloxacin triggering expression changes with higher variance from bacterial cultures having low fitness (*S. typhimurium* and *S. pneumoniae* serotype 1), compared to those with high fitness (*S. pneumoniae* serotype 23F, *E. coli* and *K. pneumoniae*) (Fig. 5b). Single-timepoint entropy was computed for the transcriptome of each of these previously unseen isolates. Importantly, with the original threshold of 2.08, which was determined during model training with data from *S. pneumoniae* in Fig. 4, fitness outcomes could be predicted for the new organisms with 100% accuracy, indicating that the single-timepoint entropy measure, which uses the least amount of data compared to other variants of entropy, is a species-independent generalizable feature for fitness outcome.

**Fig. 4 Fitness can be accurately predicted using a single time-point based definition of entropy. a** Genome-wide differential expression (indicated as log2FoldChange Antibiotic/NDC (no drug control)) shows significantly wider distributions in antibiotic-sensitive strains (wtTIGR4 and wt19F) compared to antibiotic-adapted strains in the presence of vancomycin (a cell wall synthesis inhibitor; CWSI) and rifampicin (an RNA synthesis inhibitor; RSI), respectively in a two-sided Kolmogorov–Smirnov test. **: $0.0001 < p < 0.001$; ***: $p < 0.0001$. **b** Entropy for a single time point is defined as the log-transformed variance of the distribution of differential expression across genes for a specific timepoint. **c** Single time point entropy is calculated from differential expression of all genes in experiments in the training (left panels) and test (right panels) datasets at each time point and plotted against time post-stress exposure (i.e., in the presence of antibiotics—AMX, CEF, CFT, CIP, COT, DAP, IMI, KAN, LIN, LVX, MOX, PEN, RIF, TET, TOB, VNC, or in the absence of nutrients—Glycine-GLY, Uracil-URA, Valine-VAL). Dashed red line indicates the entropy threshold (2.08) for the single-timepoint entropy predictions of fitness. The performance of the single time-point entropy-based fitness prediction (applied to all timepoints, ranging from 10′ to 240′) is shown as receiver-operator characteristic (ROC, **d**) and precision-recall (PR, **e**) curves. The area under the ROC curve is 0.79 and 0.88 for training and test sets, respectively. The area under the PR curve is 0.77 and 0.96 for training and test sets respectively. **f** Confusion matrix of single time-point entropy-based fitness prediction of the training (top panel) and test (bottom panel) datasets, highlights a good performance, but shows that there are a relatively large number of false positives. **g** Entropy values of individual experiments in the training (top) and test (bottom) sets, separated by time. Left and right panels show early (≤45 min) and late (>45 min) timepoints, respectively. It turns out that most false-positive predictions in panel **f** come from early timepoints due to a lack in transcriptional changes within the first 45′ after antibiotic exposure. In contrast, antibiotic exposure longer than 45′ (late timepoints) leads to a clear separation of high and low fitness and high accuracy in training and test data sets.

Furthermore, the entropy measurement of each strain was found to be inversely proportional to the MIC$_{CIP}$ (Fig. 5c), consistent with transcriptional disruption being proportional to stress sensitivity. The correlation between entropy and ciprofloxacin sensitivity in Fig. 5c (left panel) therefore implies that the antibiotic sensitivity of other species could be predicted from its transcriptomic entropy. To test this, entropy was calculated for *Acinetobacter baumannii* isolates that are either low (ATCC 17978) or high (LAC-4) virulence, by collecting RNA-Seq profiles after 120 min exposure to 1 μg per mL of ciprofloxacin. Using a linear regression model, the ciprofloxacin MICs of the *A. baumannii* strains were predicted to be 0.04 and 10.45 μg per mL, which are proximate to the measured MIC's of 0.07 and 8.5 μg per mL for ATCC 17978 and LAC-4, respectively (Fig. 5d; Supplementary Fig. 1D). This demonstrates that entropy is not simply a binary indicator of fitness outcomes. Even when using a single timepoint, i.e., the least amount of transcriptomic information, entropy can be applied to determine the antibiotic sensitivity level for new unseen species that were not in any training data.

To further validate the approach, data from Bhattacharyya et al.[11] was used. In this RNA-Seq dataset, susceptible and resistant strains from three species were exposed to three different antibiotics (two of which were not present in our dataset). Again, by using the entropy threshold of 2.08 (obtained above through training on the *S. pneumoniae* data) susceptible strains with low fitness are successfully separated from resistant strains with high fitness (Fig. 5e).

Finally, to explore the applicability of entropy beyond nutrient and antibiotic stress, entropy-based fitness classification was performed on a published collection of 193 *M. tuberculosis* transcription factor overexpression (TFOE) strains[39]. Upon TFOE, these strains exhibit fitness changes, ranging from severe growth defects to small growth advantages[40]. Overexpression of a single transcription factor can thereby exert stress on the bacterium that can result in different fitness outcomes. By calculating entropy from whole-genome microarray data collected from each TFOE strain, it is possible to distinguish strains based on their fitness levels at an accuracy of 0.78, using a newly trained entropy threshold for this dataset (Fig. 5f). This result compares favorably with a much more complicated approach involving the integration of each TFOE transcriptional profile into condition-specific metabolic models[39]. Overall, these data clearly highlight the strength of entropy, which has the potential to be utilized as a generalizable fitness prediction method for both antibiotic and nonantibiotic stress, and a large variety of bacterial strains and species.

## Discussion

A major goal of this work is to determine if there is a quantifiable feature that can accurately predict bacterial fitness in an environment, independent of strain, species or the type of stress. To be generalizable, the selected feature needs to be common across species and environments. By generating a large experimental dataset and analyzing published ones, we show that such a feature exists, namely transcriptomic entropy, which quantifies the level of transcriptional disorder while a bacterium is responding to the environment. It is important to realize that entropy is not simply a measure of large magnitude changes in the transcriptome. Instead, entropy takes into account condition-specific transcriptional dependencies among genes, and quantifies the amount of independent changes. The underlying assumption is that gene expression patterns lose underlying dependencies and become more stochastic with increasing amounts of stress. The difference between simple measures of magnitude changes and more controlled measures of entropy is illustrated in Fig. 3b. We show that entropy is a flexible, and generalizable predictor of bacterial fitness in a variety of different environments, it can be used with time-course data or single-timepoint data, and can even be used to predict the MIC of an antibiotic. This study demonstrates how entropy-based predictive models can be implemented in several ways, by using different amounts of data, resulting in different types of predictions. Even using a single timepoint, it is possible to predict both fitness as a binary outcome, as well as the MIC of an antibiotic (Fig. 5d), highlighting entropy as a very flexible framework that can be adapted to different settings.

We use current gene-panel based approaches for two reasons: (1) To search for a gene-panel that would capture a general stress-response (if it exists), and thus would represent a set of genes and associated regulatory changes coordinated by the same mechanisms in response to different types of stress. The existence of such a general response has been mostly connected to the manner in which *rpoS* responds to stress in *E. coli* and a small number of other species. However, it is largely unclear which genes respond downstream of *rpoS*, whether this response is accompanied by stress-specific responses, to what extent these transcriptional changes overlap across species and in response to different types of stress[16]. Moreover, if such a general stress response exists widely across species, it is unclear whether there is any predictive information to be extracted from it. Importantly, we were unable to identify such a gene-panel within the dataset we generated for *S. pneumoniae* and other species, as well as in the published datasets we explored; (2) as a point of comparison for our entropy-based approach. This comparison highlights that an entropy-based approach yields better

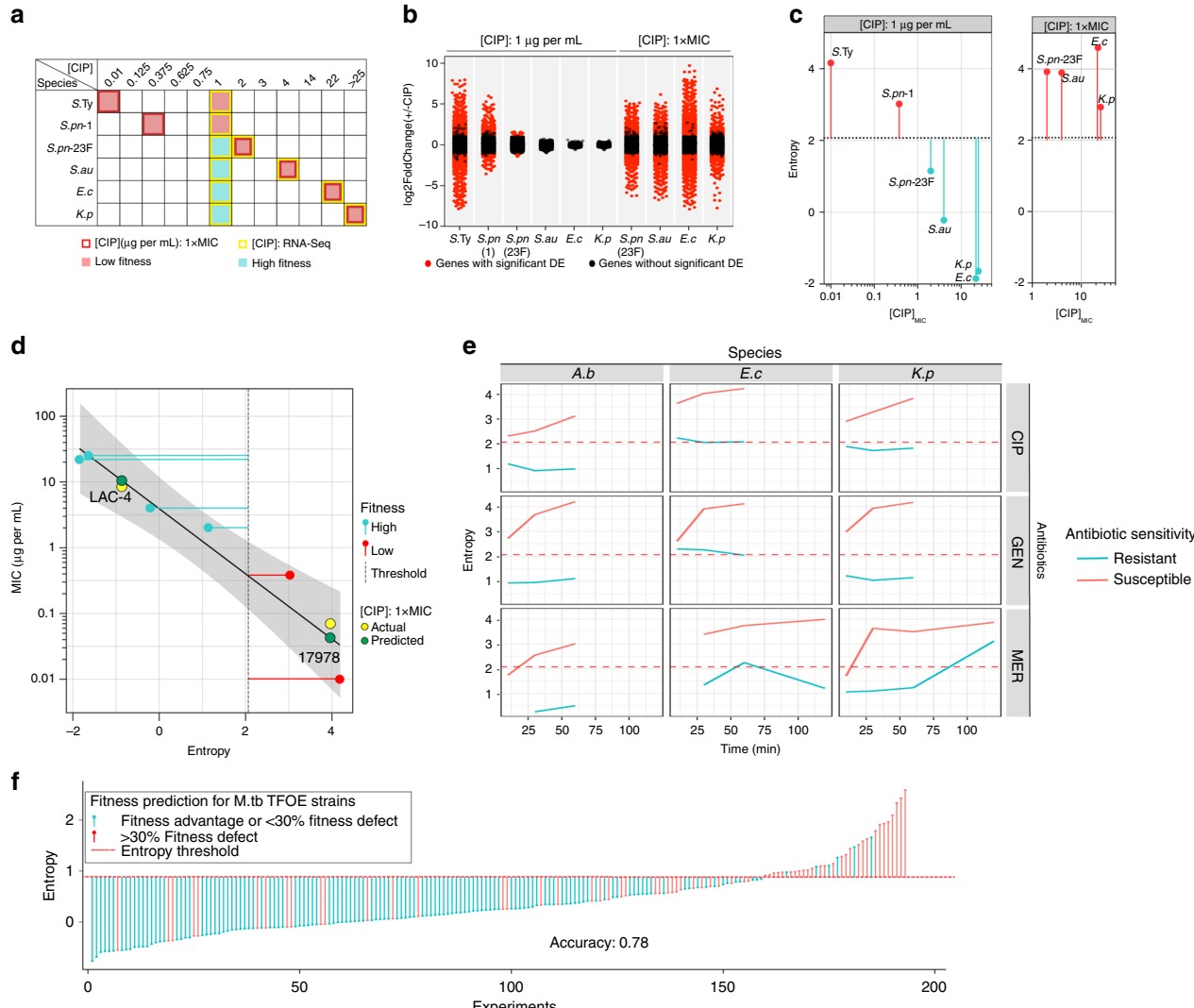

**Fig. 5 Entropy-based predictions extend to multiple species under antibiotic or regulatory stress. a** Six strains representing five species are ranked from low to high ciprofloxacin minimal inhibitory concentrations (MIC$_{CIP}$) tested by growth curve assays (Supplementary Fig. 1). The multi-species CIP RNA-Seq is performed at two CIP concentrations: (1) 1 μg per mL for all six strains corresponding to two low fitness outcomes (red squares) and four high fitness outcomes (cyan squares); (2) MIC$_{CIP}$ for strains that are insensitive to 1 μg per mL of CIP, i.e., *S. pneumoniae* serotype 23F, *S. aureus* UCSD Mn6, *E. coli* AR538, and *K. pneumoniae* AR497, corresponding to four additional low fitness outcomes. The number of genes that change in expression upon exposure to 1 μg per mL CIP (|log2FoldChange| > 1 and *p*-adj < 0.05) as well as their change in magnitude is inversely correlated to their CIP sensitivity (**b**) and their entropy (**c**). In addition, strains with MIC$_{CIP}$ higher than 1 μg per mL revert to triggering a large number of differential expression genes (**b**) and a high entropy (**c**) at their respective 1× MIC$_{CIP}$. **d** Using a linear regression model (black line; error band: 95% confidence interval for the regression), MIC's are predicted for *A. baumannii* strains ATCC 17978 and LAC-4 based on their entropy at 1 μg per mL of ciprofloxacin. The predicted (green datapoint) and measured (yellow datapoint) MIC for the two strains are highly accurate indicating that entropy can be used as a quantitative predictor. See Supplementary Fig. 1D for MIC determination for *A. baumannii* ATCC17978 and LAC-4. **e** Further validation of the generalizability of the single time-point entropy approach on published expression data[11]. The universal entropy threshold of 2.08 trained on our *S. pneumoniae* data, was successfully used to predict fitness outcomes of susceptible and resistant strains from three species in the presence of three different antibiotics. Importantly, six of the species-antibiotic combinations (GEN-*A.b/E.c/K.p* and MER-*A.b/E.c/K.p*) were not present in our datasets, which highlights the universality and generalizability of the entropy based approach. GEN gentamicin, MER meropenem. *A.b A. baumannii, E.c E.coli, K.p K. pneumoniae*. **f** Entropy calculated from transcriptional profiles of 193 *M. tuberculosis* transcription factor overexpression (TFOE) strains from reference[35] separates strains with a >30% fitness defect upon TFOE induction (red) from strains with a fitness advantage or <30% fitness defect upon induction (cyan). At the threshold of 0.71 (red dotted line), fitness outcomes are correctly predicted at an accuracy of 0.78.

performance than a gene-panel based approach (Supplementary Data 6), and has at least three additional advantages over existing gene-panel approaches: (a) It is independent of specific genes, whereas gene-panels focus entirely on specific genes. This might lead researchers to interpret genes present in a particular panel as those most relevant to the stress response. However, caution should be taken in the interpretation of these gene panels,

because it turns out that the genes that appear in these panels are strongly influenced by model parameters (λ) and input data (Fig. 1). (b) An entropy-based method has few (at most 2) parameters, and therefore does not risk overfitting (unlike gene-based approaches, where there is at least one parameter per each transcriptionally measured gene). (c) The entropy method generalizes across different antibiotic and non-antibiotic conditions,

and across different species. This is not the case for gene-panel based methods, which can only make predictions on the same conditions as the data they were trained on (i.e., one model is predictive for a specific species and a specific antibiotic). And even though a gene-panel may only use expression of a limited number of genes to predict fitness, and may therefore seem to be relatively easy to implement in a clinical setting, each new antibiotic-species combination requires the collection of an entirely new training dataset. This makes gene-panel approaches costly. Although in this paper we focus mainly on accuracy of fitness predictions, there are additional biological insights to be gleaned from the data presented in this work. For instance, the inverse covariance matrix from Eq. (1) represents a network that reveals regulatory interactions among genes. The covariance network inference using graphical-lasso regularization presented here is to the best of our knowledge an improvement upon other methods (e.g., WGCNA[41]), which will be explored in depth in future work. Thus, it is possible that the networks generated in this work will be applicable in other ways, e.g., in the identification of novel regulators, their targets, or the prediction of transcriptional changes that follow a perturbation.

By demonstrating the feasibility of predictions of fitness outcomes and antibiotic sensitivity, we envision several possibilities of integrating entropy-based predictions in a clinical diagnostic setting. Currently, AST is often performed using culture-based methods. These methods may take days and even weeks for slow-growing species such as *M. tuberculosis*[42], delaying diagnosis and treatment in clinical settings. Therefore, it is desirable to be able to predict the fitness outcome of such slow-growing species as early as possible, for instance using RNA expression data. Another potential application of our entropy-based fitness predictions is monitoring an active infection in vivo. Performing transcriptome profiling and predicting the fitness of the infectious agent directly in its host environment would allow for monitoring of disease progression, and determining if and when treatment is necessary. Simultaneously profiling the pathogen and the host using dual RNA-Seq[43,44], and predicting the fitness of both could also be valuable in assessing the state and progression of an infection.

Admittedly, direct implementation of RNA-Seq in diagnostic tests might not (yet) be practical, as RNA-Seq experiments still remain relatively expensive, labor-intensive and time-consuming. In particular, time-course experiments such as those included in this study increase in cost linearly with an increasing number of time points. However, the advances in technology are likely to reduce cost much more drastically than a linear model, as is observed for many sequencing approaches. To implement temporal entropy, it is important to recognize that more timepoints will yield better results. However, even two timepoints gives robust results. The most economic approach would clearly be the single-timepoint model, which has comparable performance to the temporal models, with the only disadvantage that it lacks possible insights that could be gleaned from the covariance networks temporal entropy is based on. With the advent of real-time sequencing technologies, such as Nanopore, the speed of data collection may soon be improved significantly. In addition, a transcriptome can be sub sampled by monitoring conserved genes across species. In this scenario, transcriptional entropy can be obtained via more economical gene expression technologies, such as NanoString nCounter[45] or the Luminex platform[46]. To conclude, we present an approach that uses entropy to predicting fitness independently of gene-identity, gene-function, and type of stress. This approach can be applied as a fundamental building block for generalizable predictors of fitness and MICs for Gram-positive and negative species alike, and thereby possibly improve clinical decision-making.

## Methods

**Bacterial strains, culture media, and growth curve assays.** *S. pneumoniae* strain TIGR4 (T4; NC_003028.3) is a serotype four strain originally isolated from a Norwegian patient[47,48], Taiwan-19F (19F; NC_012469.1) is a multi-drug resistant strain[49,50] and D39 (NC_008533) is a commonly used serotype 2 strain originally isolated from a patient about 90 years ago[51]. Strain PG1 and PG19 were isolated from adults with pneumococcal bacteremia infection and included in the Pneumococcal Bacteremia Collection Nijmegen (PBCN)[52]. All *S. pneumoniae* gene numbers refer to the T4 genome. Correspondence between homologous genes among *S. pneumoniae* strains and gene function annotations are described in Supplementary Data 1. *Escherichia coli* strain AR538, *Klebsiella pneumoniae* strain AR497 and *Salmonella enterica subsp* Typhimurium strain AR635 were clinical isolates obtained from the Center of Disease Control (CDC). *Staphylococcus aureus* strain MN6 was kindly provided by George Sakoulas (Center of Immunity, Infection & Inflammation, UCSD School of Medicine). Unless otherwise specified, *S. pneumoniae* strains were cultivated in Todd Hewitt medium with 5% yeast extract (THY) with 5 µL per mL oxyrase (Oxyrase, Inc) or on sheep's blood agar plates (Northeastern Laboratories) at 37 °C with 5% $CO_2$. *A. baumannii, E. coli. K. pneumoniae, S. aureus*, and *S.* Typhimurium were cultured in Mueller Hinton broth II (Sigma) at 37 °C with 220 rpm constant shaking. RNA-Seq experiments of *S. pneumoniae* under nutrient-depletion and antibiotic conditions were performed in semidefined minimal medium (SDMM)[20]. RNA-Seq experiments for *A. baumannii, S. typhimurium, E. coli. K. pneumoniae*, and *S. aureus* were performed in Mueller Hinton broth II. Single strain growth assays were performed three times using 96-well plates by taking $OD_{600}$ measurements on a Tecan Infinite 200 PRO plate reader.

**Temporal RNA-Seq sample collection, preparation and analysis.** In nutrient RNA-Seq experiments, T4, D39, and adapted D39 were collected at 30 and 90 min after depletion of D39-essential nutrients. In the training set antibiotic RNA-Seq experiments, wild-type and adapted T4 or 19F were collected at 10, 20, 30, 45, 60, 90, 120 min post-vancomycin, rifampicin or penicillin treatment. Additional time points at 150, 180, 210, and 240 min were collected in levofloxacin and kanamycin experiments due to the slower transcriptional response. In the test set antibiotic RNA-Seq experiments, wild-type T4 and 19F were collected at 30 and 120 min post-cefepime, ciprofloxacin, daptomycin, or tetracycline treatment. Ciprofloxacin-adapted T4 and 19F were collected at 30 and 120 min post-ciprofloxacin treatment. T4 was collected at 30 and 120 min post-amoxicillin, ceftriaxone, imipenem, linezolid, moxifloxacin, or tobramycin treatment. Wild-type strains were exposed to 1× MIC antibiotics; antibiotic-adapted strains were exposed to 1× MIC (i.e., same concentration as wild-type) and 1.5-2× MIC of the respective antibiotic. Cell pellets were collected by centrifugation at 4000 rpm at 4 °C and snap frozen and stored at −80 °C until RNA isolation with the RNeasy Mini Kit (Qiagen). Totally, 400 ng of total RNA from each sample was used for generating cDNA libraries following the RNAtag-Seq protocol[53] as previously described[17]. PCR amplified cDNA libraries were sequenced on an Illumina NextSeq500 generating a high sequencing depth of ~7.5 million reads per sample[54]. Raw sequencing data were converted to fastq files using the bcl2fastq software (v2.19, Illumina BaseSpace). RNA-Seq data was processed using an in-house developed analysis pipeline. In brief, raw reads are demultiplexed by 5′ and 3′ indices[53], trimmed to 59 base pairs, and quality filtered (96% sequence quality > Q14). Filtered reads are mapped to the corresponding reference genomes using bowtie2 (v2.2.6) with the --very-sensitive option (-D 20 –R 3 –N 0 –L 20 –i S, 1, 0.50)[55]. Mapped reads are aggregated by featureCount and DE is calculated with DESeq2 (v1.10.1)[56,57]. In each pair-wise DE comparison, significant DE is filtered based on two criteria: |log2foldchange| > 1 and adjusted *p* value (padj) < 0.05. All DE comparisons are made between the presence and absence of the antibiotic or nutrient at the same time point. The reproducibility of the transcriptomic data was confirmed by an overall high Spearman correlation across biological replicates (*R* > 0.95). Furthermore, the consistent patterns we observe in DE for the training, test and validation experiments, as well as the similarity of DE from experiments using antibiotics with the same MOA, point to the high quality and reproducibility of our dataset. NB: comparison of experiments can be done using ShinyOmics (http://bioinformatics.bc.edu/shiny/ABX).

**Experimental evolution.** D39 was used as the parental strain in nutrient-depletion evolution experiments; T4 and 19F were used as parental strains in antibiotic evolution experiments. Four replicate populations were grown in fresh chemically defined medium (CDM) with a decreasing concentration of uracil or L-Val for nutrient adaptation populations, or an increasing concentration of ciprofloxacin, cefepime, levofloxacin, kanamycin, penicillin, rifampicin, or vancomycin for antibiotic adaptation populations. Four replicate populations were serial passaged in CDM or SDMM as controls to identify background adaptations in nutrient or antibiotic adaptation experiments, respectively. When populations were adapted to their nutrient or antibiotic environment, a single colony was picked from each experiment and checked for its adaptive phenotype by growth curve experiments.

**Determination of relative MIC.** Totally, $1-5 \times 10^5$ CFU of mid-exponential bacteria in 100 µL was diluted with 100 µL of fresh medium with a single antibiotic to

achieve a final concentration gradient of cefepime (T4: 0.008–0.8 μg per mL; 19F: 0.6–2.4 μg per mL), ciprofloxacin (*S. pneumoniae* strains: 0.125–4.0 μg per mL; other species: 0.0125–25 μg per mL), daptomycin (15–55 μg per mL), levofloxacin (0.1–2 μg per mL), kanamycin (35–250 μg per mL), penicillin (T4: 0.02–0.055 μg per mL, 19 F: 1–4 μg per mL), rifampicin (0.005–0.04 μg per mL), tetracycline (T4: 4–18 μg per mL; 19F: 19–22 μg per mL); amoxicillin (0.01–0.16 μg per mL), imipenem (0.0005–0.045 μg per mL), ceftriaxone (0.0005–0.009 μg per mL), linezolid (0.05–0.65 μg per mL), tobramycin (35–255 μg per mL), cotrimoxazole (0.5–7.5 μg per mL); moxifloxacin (0.05–0.70 μg per mL), and vancomycin (0.1–0.5 μg per mL) in 96-well plates. Each concentration was tested in triplicate. Growth was monitored on a Tecan Infinite 200 PRO plate reader at 37 °C for 16 h. MIC is determined as the lowest concentration that abolishes bacterial growth (Supplementary Fig. 1).

**Selection of a gene panel for fitness prediction**. DE data from experiments from all experimental timepoints with time ≥ 60 min were assembled in R (v3.6.2). The data were split into training and test sets as described in Supplementary Table 1, yielding a training set of 138 and a test set of 19 experiments. Genes with incomplete data (e.g., genes unique to one strain) were omitted. The DE data was then scaled such that the values for each gene had mean = 0 and variance = 1. A binomial logistic regression model was fit to the training set with glmnet v3.0–2. In order to determine the appropriate value of the regularization parameter λ, fivefold crossvalidation was performed on the training set, and mean squared error (MSE) of the crossvalidation set for each of the fivefolds was computed as a measure of classification error. The value of λ was selected to be the largest at which the MSE is within one standard deviation of the minimal MSE overall[25,26]. The heatmap of DE for this gene panel was generated using heatmaply (v1.0).

Evaluation of the gene panel's sensitivity to input data was done using another fivefold crossvalidation strategy, where for each fold, the training portion includes 80% of the original training dataset. The model was fit with the same strategy as above, selecting the best λ for each fold.

Evaluation of the gene panel's sensitivity to λ was done using the standard output of the glmnet function.

For gene panels specific to a single MOA, the training and test sets were filtered to include only experiments from that MOA. The model fitting procedure was the same for all gene panels that predict fitness. Performance statistics and visualization were done using plotmo (v3.5.6), caret (v6.0-85), PRROC (v1.3.1), and ggplot2(v3.2.1).

**PCA and trajectory clustering**. For PCA, DE (log2fold change of ±antibiotic comparisons) data from all 255 experimental conditions (per time point per antibiotic from all experiments excluding CIP-validation set with *A. baumannii*, *E. coli*, *K. pneumoniae*, *S.* Typhimurium, *S. aureus*, *S. pneumoniae* serotype 1 and 23F strains) were assembled in R (v3.6.2). The function "prcomp" was used for PCA. Timepoints of the same experiment were connected to form trajectories. Since not all experiments are on the exact same time scale (e.g., KAN experiments extend to 240 min whereas RIF experiments cover 120 min), equivalent timepoints for each experiment were determined to be $\frac{i \times t_{max}}{6}$ for $i = 1, 2, \ldots, 6$ and $t_{max}$ being the latest time point available for the corresponding experiment. If a timepoint did not correspond to an existing RNA-Seq data point, this time point was inferred by linear interpolation of the existing trajectories. To cluster these trajectories, a trajectory-distance metric between two trajectories $X$ and $Y$ is defined as the sum of Euclidean distances ("dist", on the principal component coordinates) $\Sigma_{i=1}^{6} \text{dist}(X_i, Y_i)$ of all timepoints $i$. All pairwise distances are computed for all pairs of trajectories included in the analysis (WT strains with low fitness, for PSI, DSI, CWSI, and RSI). K-means clustering in MATLAB with $K = 4$ is used on the pairwise distances to cluster the trajectories.

**Selection of a gene panel for MOA prediction**. DE (log2 fold change of drug/no drug comparison) data from all antibiotic experiments with low fitness outcome and time ≥60 min were assembled in R (v3.6.2). The data were split into training and test sets as described in Supplementary Table 1, yielding a training set of 39 and a test set of 15 experiments. Similar to the fitness gene panel data preparation, genes with incomplete data were omitted. A multinomial logistic regression model was fit to the training set with glmnet v3.0-2. The appropriate value of λ was selected using a similar crossvalidation scheme to the fitness gene panel: the largest λ at which the corssvalidation error is within 1 standard deviation of the minimal error overall.

Visualization, and evaluation of the model's performance, sensitivity to input and λ were done as described in the "Selection of gene panel for fitness prediction" section above.

**Gene set enrichment analysis**. Gene panels for *S. pneumoniae* were evaluated for enrichment of functional categories (the category annotation can be found in Supplementary Data 1), using a hypergeometric test, and Benjamini–Hochberg correction for multiple comparisons. For gene panels in Bhattacharyya et al.[11], enrichment for GO terms was evaluated using the same procedure. The GO term annotation was acquired from Uniprot.

**Quantifying entropy of transcriptomic data**. Entropy (H) for a time-course experiment is defined as in Eq. (1). The DE data for the time-course is assembles into a single matrix $S$, where columns are individual genes, and rows are different time points. The covariances across all pairs of columns (i.e., genes) is computed using the "cov" function in R (v3.6.2) to generate the covariance matrix ($\Sigma$). $\Sigma$ is then used as input for the glasso function within the glasso package (v1.10), which generates a regularized covariance matrix ($\Sigma_\rho$). Multiple values of $\rho$ are scanned between 0 and 5, and for each value of $\rho$, the error on the training set was computed. The value of $\rho$ was determined to be that which minimized error. Using this value of $\rho$, multiple values of threshold $t$ were scanned within the range of entropy values within the training set. The value of $t$ was determined to be that which maximized accuracy on the training set.

Entropy of a single timepoint ($H_{stp}$) is defined as in Eq. (2). The variance ($\sigma^2$) of the whole-transcriptome DE distribution is computed using the "var" function in R (v3.6.2). The threshold value $t$ was determined by scanning the range of $H_{stp}$ values in the training set, and finding the $t$ that maximized accuracy on this dataset.

The predictive performance of all entropy models was evaluated on both the training and test sets using caret (v6.0-85), PRROC (v1.3.1); and visualized using ggplot2(v3.2.1).

**Reporting summary**. Further information on research design is available in the Nature Research Reporting Summary linked to this article.

## Data availability
Raw RNA-Seq datasets are available at the Sequence Read Archive (BioProject accession number PRJNA542628). Differential expression data used in all main figures can be found in Supplementary Data 1. Results for gene set enrichment analysis on previously published gene panels are in Supplementary Data 2. Entropy values and predictions associated with Fig. 4 are in Supplementary Data 3. The previously published RNA-Seq dataset used in Fig. 5 is available under the BioProject accession number PRJNA518730. Gene homology across species was obtained from PATRIC [https://www.patricbrc.org/], and gene functional annotation for enrichment analysis was obtained from UniProt [https://www.uniprot.org/]. Source data are provided with this paper.

## Code availability
Custom code used in the analysis and generation of figures can be found in the GitHub repository named FitnessPrediction [https://github.com/dsurujon/FitnessPrediction].

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

## Acknowledgements

DNA sequencing was performed at the Boston College Sequencing Core. The authors wish to thank Jon Anthony for running the Aerobio sequencing analyses pipeline. This work was supported by NIH R01 AI110724 and U01 AI124302.

## Author contributions

T.v.O. devised the study. Z.Z., J.O., W.H., and T.v.O. performed the wet-lab experiments. D.S. performed the computational experiments. J.B. contributed to the key conceptual ideas. Z.Z., D.S., J.O., W.H., R.I., J.B., and T.v.O. analyzed the data. Z.Z., D.S., J.O., J.B., and T.v.O. interpreted results and wrote the paper. Z.Z., D.S., J.O., W.H., R.I., J.B., and T.v.O. approved the final paper.

## Competing interests

The authors declare no competing interests.
