## [Peer Review File · Nature Communications]

Reviewers' Comments:

Reviewer #1:

Remarks to the Author:

This paper describes the development of a methodology to predict bacterial phenotypes based on gene expression profiles. The author proposed a measure called entropy to represent fitness changes, by which they demonstrated that fitness could be predicted in various microbial species. The method for fitness prediction based on a small number of features is quite impressive, and it might contribute to medical applications. However, the presented analyses are limited, and the conclusions are not fully supported.

Major comments:

i) In the analysis shown in Fig.1D, are different sets of genes selected when different sets of training data and test data are used? How does the prediction accuracy (Fig. 1E) change when the number of genes in the gene panel is changed? Throughout the paper, the authors present the results based on a small number of conditions (e.g., training/test sets, parameters). From such presentations, it is difficult for us to evaluate the robustness of results. The authors should discuss carefully why the presented conditions are used and how they are robust against condition changes.

ii) In Fig. 2A and 2B, the authors show that there is a difference in the distribution of DE genes between the stress-sensitive and adapted strains. This result is quite natural because the adapted strains remain a similar state (probably exponentially growing state) with that before the addition of antibiotics, while the stress-sensitive strains fall into a different state with growth suppression. Is it a novel finding?

iii) The authors wrote: "Finally, stress-sensitive strains have a significantly lower KLD relative to adapted strains, indicating higher similarity of DE genes compared to the distribution of the entire genome." Here, although the detailed method for the calculation of KLD is not presented in Materials and Method, I guess (based on Fig. 2A iv) that the difference in the ratio of genes in different functional categories is evaluated by KLD. If it is the case, the result can be influenced by the difference in the number of DE genes between stress-sensitive and adapted strains. In the adapted strains, the number of DE genes seems much smaller than the stress-sensitive strains (Fig. 2A ii). Thus, in the adapted strains, the ratio of the number of genes in different categories can be fluctuated due to the small number of DE genes, resulting in larger KLD. The authors should present the number of DE genes used for the calculation of KLD, and correctly evaluate the effect of different numbers of DE gene between stress-sensitive and adaptive strains. For example, for the stress-sensitive strains, how does the distribution of KLD (Fig. 2C) change when the same number of DE genes with the adapted strains are randomly sampled?

iv) In Fig. 3, the authors show that the difference in the fitness (growth activity) can be predicted by the 10-gene panel. It seems an easy task. There are several previous studies demonstrated that we can identify the global expression pattern correlated to growth rate, e.g., <https://journals.plos.org/plosone/article?id=10.1371/journal.pone.0153344>
<https://journals.aps.org/prx/abstract/10.1103/PhysRevX.5.011014>
These studies indicated that we can find genes whose expression levels are highly correlated to the cellular growth, by which the analysis in Fig. 3 is possible. Here, I could not find any novelty.

v) Again, how do the genes selected in Fig. 3A and B depend on training/test sets?

vi) The concept of "entropy" to represent macroscopic changes in phenotype is interesting. However, it seems equivalent to the concept that "Decrease of growth activity from the standard condition is positively correlated to the overall changes in expression profile", which was already presented by previous studies such as those presented above. What is the novelty here?

vi) In the analysis shown in Fig. 4, is "model 3" best to predict fitness change? The prediction accuracy of model 3 is almost the same as model 1. If the conclusion is that model 3 is superior to other models, the authors should verify it by appropriate statistical analysis. If it is the case, what is the conclusion by comparing these models 1 to 3?

vii) In the analysis shown in Fig. 7, the authors should present how the accuracy changes by changing features used for SVM analysis. For example, does SVM using only KLD or entropy (or both) as the features result in lower accuracy? What happens when all the features are used? In addition, what happens when training/test sets are changed for the feature selection? The same feature sets are always selected independently of training/test sets? In this paper, the authors tend to present only one representative result, but results by changing parameters and conditions should also be presented with appropriate statistical analysis to facilitate a better understanding of the generality and robustness of the results.

viii) When we consider possible applications of this entropy-based method, the advantage remains unclear. Even when the fitness can be predicted by this method, certainly the transcriptome analysis is expensive compared with direct observation of the fitness (e.g., MIC for antibiotics). If such direct measurement is difficult for some reason, measuring a small number of panel-genes is much easier. Of course, for each microorganism species, seeking appropriate panel-genes is necessary, but it is possible. After finding such panel genes, the experimental cost can be significantly smaller than the entropy-based method proposed in this study. The authors should discuss in what situation this method can be best for analyzing bacterial phenotype.

Minor comments:

i) In Fig. 1C, I guess the order of vertical labels is incorrect.

ii) In page 5, line 129, the gene code of *clpP* should be presented (*sp_0338*); otherwise, the readers cannot identify it in Fig. 1D.

iii) The authors explain why two strains T4 and T19 are used in this study. What are the differences in the phenotype of these strains? What results are expected by the authors? Is it for checking reproducibility?

iv) The detailed explanation of the D39 strain is helpful in understanding why this strain is used in this study.

v) The vertical axis in Fig. 2A i should be log-scale. The most important information from this growth curve is whether the adapted strains are in an exponential growth phase at the sampling time point (90 min.),

vi) The detailed explanation and definition of KLD should be added in Materials and Methods.

Reviewer #2:

Remarks to the Author:

van Opijnen and colleagues conducted a systematic study to determine whether there exists a universal quantifiable feature in bacterial response pattern that is predictive of fitness, irrespective of stress type and species. They developed, optimized and examined a set of predictive models based on experimental evolution experiments and found that the transcriptome entropy, a measure of systems perturbation in face of external stress is predictive of fitness under nutrient and antimicrobial stresses. They also presented a gene-panel based modelling to predict the mechanism of action of antimicrobials with a high accuracy.

The study is timely and takes an innovative approach, in terms of both experimental design and analytical framework. Despite a large literature on diagnostic tests, in particular in the context of genetics of AMR, there has not been much of effort to examine dynamics of transcriptome and this study provides considerable insights into the issue. However, I have significant concerns about the implementation of Machine Learning (ML) methods and justifications of test design and therefore cannot recommend the manuscript for publication in its current format.

A major comment/concern is concerned with details of the implementation machine learning methods throughout the paper. In spite of the importance of the methods in obtaining the final conclusions and results, the predictive methods are not adequately detailed. As a result, it is impossible to assess whether results are reproducible and robust. In the methods section, only a few lines are dedicated to describe the ML methods.

In particular the methods section needs to include:

1- Method justification: the authors have used SVM without justifying the choice of the method. In two other analyses, logistic regression was used. I wonder if the response feature is sufficiently separated for an SVM to work properly and also whether SVM outperforms simpler logistic regression.

2- There is no mention of validation dataset throughout the paper and as a result it is unclear whether the model is tested on a completely unseen test dataset.

3- The description of cross-validation to prove that the markers are robust and overfitting has been avoided is missing. Authors need to provide the error rate for test and training data sets. Also, the model ideally should be tested on different held-out datasets to show that feature importance values are robust.

4- Hyperparameter tuning and details of optimized model are absent. Also given the limited number of samples in the test dataset for predicting MOA, as indicated in Figure 1, I wonder if authors have introduced any regularization in the SVM methods. This can include L2 and C regularization and using kernel methods.

5- To report the performance of a diagnostic tool, accuracy is not informative and other prediction metrics, e.g. AUC, specificity, recall and precision, should be used.

6- The choice of six genes in Line 333 is not justified. Also feature importance analysis requires an examination of co-linearity between features, which is not provided. It is unclear why authors decided to reduce predictor features without assessing the extent of overfitting when all features are used.

The choice of 8 genes (Line 122), later 10 genes (Line 193) in the gene panel seem arbitrary and have been weakly justified. I expect the authors to conduct a sensitivity analysis to demonstrate how inclusion of more gene will affect the predictive power and whether mining the input data into few genes has not resulted in a significant information loss. By doing so, authors can show that their threshold/cut-off choice for markers numbers is objective.

Given that authors have studied only a few representative strains, I think the conclusion on the applicability of prediction power of entropy to other conditions/species is a bit far-fetched.

Importantly, the effect of genetic background is not discussed in the limitations. Also, from supplemental table S1, it seems the method has been examined only on few strains from other species and therefore the results are not generalizable to the population level.

Furthermore, in the limitation section, authors should mention that the entropy metric has been tried in a controlled experimental evolution condition in the lab, which does not necessarily represent natural and clinical conditions.

From a clinical point of view, a diagnostic test should not only have a high accuracy but provide a mechanistic understanding of the underlying features. The link between entropy and stress does not seem to have the latter or at least this is not adequately mentioned in discussion.

Other comments:

Lines 194-202 include descriptive and inconclusive results. The section looks a bit incomplete and can be either shortened or left out.

Line 209 ESKAPE should be defined.

Line 456 Support Vector Machine should be capitalized.

Lines 163-173 can be moved to Methods.

Line 132-140 The functional enrichment analysis is unconvincing and sounds like cherry-picking. It is unclear whether or not the occurrence of the genes merely reflects the underlying distribution of gene functions.

Lines 231-250 can be significantly shortened.

Lines 82-85 sound unclear and should be rephrased/elaborated, given the importance of the message.

Reviewer #3:

Remarks to the Author:

In this work, the authors aim to construct models to classify bacterial strains as sensitive or resistant to antibiotics and predict their MIC (Minimal Inhibitory Concentration) using global statistics of their transcription response to antibiotics. First, the authors demonstrate the limited predictive capabilities of specific-gene models. Second, they demonstrate that a variety of environmental stresses induce widespread transcriptional changes in WT (stressed) versus pre-adapted (unstressed) populations, and thus, can be used to classify populations and predict MIC. In particular, they introduce the notion of predictions based on "entropy" - the degree to which transcriptional changes occur due to antibiotic treatment. Third, they incorporated additional data from Tn-Seq and showed it can be used to improve predictions.

Major comments:

1. Overall, I have difficulty verbalizing the main contribution and novelty of this paper. I am worried (but not entirely sure) that the paper takes simple notions and wraps them in vague sexy terms. If I try to peel away the big words of "entropy" etc, the primary biological statement that emerges is rather simple and expected: the antibiotics cause more stress to the antibiotic sensitive cells than to the antibiotic resistant cells, leading to vast differences in their transcriptional response. This observation does not sound particularly surprising or novel to me. The claim that the authors discovered a new general signature of bacterial stress response therefore seems overblown. For a number of bacterial species, it is known that antibiotic exposure leads to genome-wide changes in gene expression due to both generalized and specific stress responses (e.g., Gottesman, 2019). I view "entropy" as a way to quantify this stress response, but the underlying phenomenon is not new. I want to be careful here: I may be missing something profound and if so I would be glad to be corrected.

2. The primary application – transcriptome-based diagnostics – doesn't seem practical, at least not in the near future. And if it does turn practical, I am most certain that an approach based on ML model trained on the gene-specific transcriptional response (rather than global statistics) of several species to different antibiotics will do much better. Even if it does show superiority, I would also appreciate a more detailed discussion of how the authors envision their method being used in the clinic, since I have a hard time envisioning its practical utility. Specifically, the proposed approach requires measuring a full transcriptome, which seems more expensive and more time-

consuming than simply measuring the MIC.

Minor comments:

1. The term "entropy" is more confusing than helpful to me. As I understand it, "entropy" simply quantifies the variance in the $\log_2(\text{gene expression} \pm \text{drug})$ distribution. However, it was difficult to parse this from the text. Furthermore, given the variety of meanings that "entropy" can take, the term is easily misinterpreted, depending on the reader's background. Perhaps the term "stress-induced transcriptional change" would be more transparent.

2. Fig. 1C: One of the axes is flipped.

3. In multiple cases, the same data is shown via two different representations, but to support the same overall message. For example, Fig. 2B and 2C are redundant. Removing these panels will help to streamline the paper.

We thank all reviewers for their comments and suggestions. We have made substantial changes to the main text, figures and supplemental materials, we've added important new analyses, analyzed data from other publications and added a large amount of new experimental data. Several of the most important changes include:

1. We realized that the main point of our manuscript was buried among a number of different points we tried to make and multiple models to support these points. Therefore, the main point may have not been entirely clear. The novelty of this work is the use of entropy to quantify disorder in a transcriptome. We carefully define and quantify disorder on temporal data, using a well-established statistical concept. We acknowledge that a similar phenomenon can be observed in ours as well as previously published transcriptome data, which is that (at a single timepoint) the number of differentially expressed genes, and the variance in magnitude of changes in expression is larger in stress-susceptible strains than stress-resistant ones. While this observation may seem intuitive, there is no formal way to quantify it for both temporal and single timepoint data, and there is no approach available that can take this observation and turn it into predictions such as fitness.

In the revised manuscript we now show in detail that entropy is a measure of disorder, which takes into account dependencies in gene expression patterns, and not simply large changes in the transcriptome. We have adjusted the language and figures to drive this point home, and to differentiate entropy from more vague interpretations that conflate it with simply magnitude changes. We also carefully and concretely define entropy in the main manuscript, and not just the supplemental material, in order to avoid any misunderstanding.

2. Centering on the concept of entropy, we have constructed a predictive method that enables generalizable predictions on bacterial fitness. Adjustments have been made throughout the text to clarify that we do not claim that the phenomenon of large transcriptional changes is a new discovery. We have also clarified that entropy is a nuanced way of quantifying disorder, and the novelty in our study is the implementation, use and generalizability of transcriptomic entropy.
3. Similar to the original manuscript we start off with generating several gene-panel based models that can be used for different predictions. The reason for this was and remains as a point of comparison for entropy. The reviewers pointed out several issues with the gene-panel based approaches, with which we agree. Here, in this new version of the manuscript we have taken the reviewers' comments and included much more detailed statistical analyses on the gene-panel approach. With the new results we have generated, we are now able to show and discuss in more detail the limitations of gene-panel based models. Moreover, we are able to better use them as a point of comparison with our entropy-based models and highlight in detail the advantages of entropy including its generalizability. In short, we emphasize that the gene-panel based models we present are included as a point of comparison for entropy. By showing their lack of generalizability, and their sensitivity to training data we highlight the limitations of gene-panel based approaches.
4. We have added 7 more antibiotic conditions to our experimental dataset, bringing the total number of antibiotics tested to 16. The new antibiotic datasets are all included in the test set, which is never included during training of any model in this manuscript. Thus, we are able to evaluate predictive performance of all models presented on a more diverse set of previously unseen conditions. Additionally, we include detailed feature selection analyses for the multiple-feature models (i.e. non-entropy models), and used the temporal entropy models on all experimental conditions (thereby enabling testing these models on a test set, and showing superior performance to gene-panels).

5. We have removed the complex feature classifier from the manuscript (formerly Figure 7), in order to streamline our argument for entropy-based models. We have included additional analyses suggested by the reviewers on this model, as part of this rebuttal letter. Even with the additional improvements, we found the complex feature classifier no better in terms of performance, and it requires additional data (e.g. Tn-Seq, functional category information on genes) for the prediction of fitness. In line with the 1st point above, we have decided that this model was superfluous, and distracted from our main point, which is entropy being a successful single-feature predictor of fitness.
6. We have added multiple paragraphs throughout the manuscript where we specifically point-out and/or discuss certain points that were raised by the reviewers. This includes; a) several sections where we discuss the possibility of the existence of a ‘general response’, which we did not find evidence for; b) the possibility that entropy could at one point be used to improve diagnostics, even though this would require improvements in technology; c) multiple sections (and figure panels) where we try to clearly explain and highlight what entropy is and what it is not, and we stress that entropy is not our invention but merely a reimagination of a classical statistical concept and its implementation in biology.

Below we address all the reviewers’ comments in detail, we list all the changes that have been made in the manuscript and we include several further analyses regarding specific points raised by the reviewers.

Point-by-point response

Reviewer #1 (Remarks to the Author):

This paper describes the development of a methodology to predict bacterial phenotypes based on gene expression profiles. The author proposed a measure called entropy to represent fitness changes, by which they demonstrated that fitness could be predicted in various microbial species. The method for fitness prediction based on a small number of features is quite impressive, and it might contribute to medical applications. However, the presented analyses are limited, and the conclusions are not fully supported.

Reviewer 1 – Comment 1:

In the analysis shown in Fig.1D, are different sets of genes selected when different sets of training data and test data are used? How does the prediction accuracy (Fig. 1E) change when the number of genes in the gene panel is changed? Throughout the paper, the authors present the results based on a small number of conditions (e.g., training/test sets, parameters). From such presentations, it is difficult for us to evaluate the robustness of results. The authors should discuss carefully why the presented conditions are used and how they are robust against condition changes.

RESPONSE:

Originally, we generated gene-panels for the prediction of fitness (similar to existing approaches) and of the mechanism of action (MOA) of a drug, which is, to the best of our knowledge, the first gene-panel for MOA prediction. The figure and analysis referred to in this comment pertain to the gene-panel that predicts MOA. We use a regression model with regularization to limit the number of features (i.e. genes). Previously, we had set the regularization strength at such a level that yielded no more than 10 genes in the panel. We have now revised this analysis. In the new Supplemental

Figure 4C, we show how the regression model for the prediction of MOA changes with different sets of input data, by showing that the coefficients of the genes that are selected change drastically when different subsets of the data are used to train the same type of regression model.

Furthermore we have included:

1. How the coefficients of individual genes are sensitive to the regularization strength in Supplemental Figure 4B;
- 2) A crossvalidation analysis (Supplemental Figure 4A) that serves two purposes: a) it shows how prediction performance changes with different numbers of features used, and b) it determines the regularization strength/number of genes automatically, in an unbiased fashion, using well-established methods(1,2);
- 3) The same analysis for the gene-panel for fitness prediction (revised Figure 1).

From these new results, we conclude that gene-panels (especially for fitness prediction) are not robust to changes in model parameters such as regularization strength, and input data, since different sets of genes are selected under different settings.

Reviewer 1 – Comment 2:

In Fig. 2A and 2B, the authors show that there is a difference in the distribution of DE genes between the stress-sensitive and adapted strains. This result is quite natural because the adapted strains remain a similar state (probably exponentially growing state) with that before the addition of antibiotics, while the stress-sensitive strains fall into a different state with growth suppression. Is it a novel finding?

RESPONSE:

We indeed observe changes in differential expression in our own data, which can also be observed in other published datasets. The effect on growth cannot be confidently observed in our experiments because the stress exposure is relatively short since we aim to make assessments about fitness before this can be clearly observed from the growth state. More importantly, the large changes in DE that can be seen in Figure 2 in the former manuscript (now Figure 4A) is not the main contribution of our manuscript, and we have made substantial changes throughout to rectify this misunderstanding. We now present these DE distributions together with the single-timepoint variant of entropy in Figure 4.

We acknowledge, in lines 90-94 in the introduction and 262-267 in the results, that similar expression changes can be observed in previous studies. We stress that the fact that this can be observed in multiple species and conditions strengthens our case, and that it could thus be a basis for a universal predictor of fitness.

We have added a section in the results (267-281) to describe how entropy builds upon this aforementioned observation, that it quantifies disorder in a very specific way, and that it can be used as a single feature that predicts fitness. Quantifying disorder in the transcriptome through entropy indeed is an important advancement over making observations on DE. Furthermore, we include a new figure (Figure 3B) and the following statement in the discussion section to differentiate and stress that entropy is different from simply large magnitude DE changes:

Lines 444-449: “It is important to realize that entropy is not simply a measure of large magnitude changes in the transcriptome. Instead, entropy takes into account condition-specific transcriptional dependencies among genes, and quantifies the amount of independent changes. The underlying assumption is that gene expression patterns lose underlying dependencies and become more stochastic with increasing amounts of stress. The difference between simple measures of magnitude changes and more controlled measures of entropy is illustrated in Figure 3B.”

Reviewer 1 – Comment 3:

The authors wrote: “Finally, stress-sensitive strains have a significantly lower KLD relative to adapted strains, indicating higher similarity of DE genes compared to the distribution of the entire genome.” Here, although the detailed method for the calculation of KLD is not presented in Materials and Method, I guess (based on Fig. 2A iv) that the difference in the ratio of genes in different functional categories is evaluated by KLD. If it is the case, the result can be influenced by the difference in the number of DE genes between stress-sensitive and adapted strains. In the adapted strains, the number of DE genes seems much smaller than the stress-sensitive strains (Fig. 2A ii). Thus, in the adapted strains, the ratio of the number of genes in different categories can be fluctuated due to the small number of DE genes, resulting in larger KLD. The authors should present the number of DE genes used for the calculation of KLD, and correctly evaluate the effect of different numbers of DE gene between stress-sensitive and adaptive strains. For example, for the stress-sensitive strains, how does the distribution of KLD (Fig. 2C) change when the same number of DE genes with the adapted strains are randomly sampled?

RESPONSE:

Initially, we had hypothesized that an adapted strain would trigger a more “targeted” response, where the functional distribution of the DE genes would be less random than the wildtype or stress-sensitive strains. We had compared whether the DE genes’ categories were similar to the category distribution of all the genes in the genome using KLD. We acknowledge that this could be influenced by the total number of DE genes in an experiment. Therefore, we re-did this analysis, with the reviewer’s suggestion. In order to test whether KLD is impacted by the number of DEGs in an experiment, we compared the KLD of each group (high and low fitness) to randomized samples with the same numbers of DEGs. In order to minimize the effect of potential outliers, we considered the mean KLD of 10 randomized samples for each experiment. We observed that KLD is indeed affected by the number of DEGs. Both high and low fitness groups had a higher KLD than their randomized controls (see figure and table below). We also added two other metrics (L1 distance and Mutual information) to compare DEG category distribution to that of the genome, in order to see whether the outcome was dependent on the metric used. For all 3 metrics, both high and low fitness cases are significantly different than random. Therefore, we remove the function distribution analysis from the manuscript and no longer claim that the category distribution of high fitness cases are less similar to the genome compared to the low fitness cases.

Figure: KLD, L1D, Mutual information (MI) of DEGs from real experiments, compared to randomized controls. The panels within each figure represent low fitness (0) and high fitness (1) cases.

To test whether the high fitness or low fitness cases were significantly different than their randomized controls, a paired Wilcoxon rank test was performed (see Table below). For all metrics, both high fitness and low fitness cases are significantly different than the randomized controls.

Metric	Fitness	p-value
KLD	LOW	5.415e-13
KLD	HIGH	0.0002953
L1D	LOW	7.134e-06
L1D	HIGH	6.6e-05
MI	LOW	9.306e-08
MI	HIGH	0.0001882

We define KLD of a set of DEGs as

$$KLD(P_{DEG}|P_G) = \sum_{x \in \mathcal{X}} P_{DEG}(x) \log \frac{P_{DEG}(x)}{P_G(x)} \quad (1)$$

Where $P_G(x)$ is the probability distribution of gene categories of the whole genome, $P_{DEG}(x)$ is the probability distribution of all DEGs in an experiment, and \mathcal{X} is the set of all categories.

L1D: This is a measure of dissimilarity of the counts of genes in each category. We define L1D as

$$L1D(DEG|G) = \sum_{x \in \mathcal{X}} |DEG(x) - G(x)| \quad (2)$$

Where $G(x)$ is the number of genes belonging to each category x on the whole genome, $DEG(x)$ is the number of genes belonging to each category for all DEGs in an experiment.

MI: This measure quantifies how much information can be obtained about one probability distribution from another. Higher MI indicates higher similarity between two probability distributions.

$$MI(P_{DEG}|P_G) = \sum_{x,y \in \mathcal{X}} P_{DEG,G}(x,y) \log \frac{P_{DEG,G}(x,y)}{P_{DEG}(x)P_G(y)} \quad (3)$$

Where $P_{DEG,G}(x,y)$ is the joint probability distribution of DEG and G .

Reviewer 1 – Comment 4:

In Fig. 3, the authors show that the difference in the fitness (growth activity) can be predicted by the 10-gene panel. It seems an easy task. There are several previous studies demonstrated that we can identify the global expression pattern correlated to growth rate, e.g., <https://journals.plos.org/plosone/article?id=10.1371/journal.pone.0153344> <https://journals.aps.org/prx/abstract/10.1103/PhysRevX.5.011014>

These studies indicated that we can find genes whose expression levels are highly correlated to the cellular growth, by which the analysis in Fig. 3 is possible. Here, I could not find any novelty.

RESPONSE:

In the previous Figure 3 (revised Figure 1) we present a gene-panel that can predict fitness. Our goal in presenting this approach is to provide a point of comparison for entropy, as gene panels have been used for predictions of growth rate (as the reviewer notes) or antibiotic sensitivity in previous literature. We have adjusted the language throughout the text to make it clear that the

gene-panel approach is not novel, and it is not the approach we advocate for. We merely use it as the existing approach for fitness predictions, and show the ways in which entropy serves this purpose better. We stress, however, that the gene-panel we present is condition-agnostic, meaning it has been trained on data coming from multiple antibiotic and non-antibiotic conditions, and tested on multiple antibiotic conditions. The existing literature presents gene panels for specific antibiotics or specific datasets, which is a crucial limitation. With the panel we present, we have attempted to extend the current approach to be non-specific to one condition, but we still find limitations of the gene-panel approach. Mainly, they are affected by the input data, model parameters, and do not generalize to new species due to a lack of homology. In lines 179-197 we highlight these points.

To go into a bit more detail with respect to the two papers referenced by the reviewer:

1. Barenholz et al.(3) identify a set of proteins whose relative abundance in the whole proteome is linearly correlated with growth rate. Similar to existing gene-panel approaches (that we also acknowledge), this approach used the abundance of specific sets of proteins - these protein sets were identified for two datasets, and crucially, they are not the same protein sets. This further supports our argument that gene/protein-panel based approaches are highly influenced by the datasets that are fed into them for training, and this is a limitation in their generalizability. With the added analyses in our revised manuscript of the sensitivity of gene-panel based approaches to input data and model parameters, we also show the drawbacks of using them. We have added a detailed discussion on the limitations of gene-panel based approaches in lines 466-478.

2. Kaneko et al.(4) present a theoretical explanation for how after bacteria have recovered from a stress and reverted to steady-growth state, gene expression can be constrained. In Fig S2, they show there is a relationship between variance across differential gene expression and reduction in growth rate. This is somewhat in line with the single-timepoint variant of entropy, however it is not further explored, developed and/or formulated in such a way that it can be used for fitness predictions as we have done. We have now cited their study in the revised manuscript and indicate that our proposed entropy-based approach in fitness prediction is partially building on observations from previous studies.

Importantly, we re-iterate that the observation of smaller overall magnitude changes associating with increased fitness is not the main contribution of our manuscript. In our work, we present a methodology that uses a similar principle as well-defined predictive models, and importantly, take into account temporal dependencies among genes, which sets our work apart from others.

Reviewer 1 – Comment 5:

Again, how do the genes selected in Fig. 3A and B depend on training/test sets?

RESPONSE:

In the previous Figure 3 (revised Figure 1) we present a gene-panel, generated using a logistic regression model. Previously we had presented a model where the regularization parameter lambda was selected such that no more than 10 genes were in the panel. We have now updated this section with analyses that better demonstrate the limitations of gene-panel based approaches. Similar to the gene panel for MOA prediction, we have included the following analyses:

1. Crossvalidation analysis to automatically determine the genes included in the regression model as features (new Figure 1B). This analysis also revealed that the model performance is not changed drastically when different values for lambda (smaller than the selected value), or different subsets of the training data are used.

2. The new Figure 1D shows that changing the data in the training set affects the coefficients of each gene drastically, which makes it difficult to interpret the selected genes in terms of their relevance to the stress
3. The new Figure 1E shows that the genes selected, and their coefficients also depend on the regularization strength, which directly affects the number of genes in the model. (we include a discussion of this issue and the previous one in lines 467-473)

By addressing this comment, we have shown the gene panel approach to be sensitive to both training data, and the number of features/regularization strength. This important issue with gene-panel approaches is now presented in lines 179-191.

Reviewer 1 – Comment 6:

The concept of “entropy” to represent macroscopic changes in phenotype is interesting. However, it seems equivalent to the concept that “Decrease of growth activity from the standard condition is positively correlated to the overall changes in expression profile”, which was already presented by previous studies such as those presented above. What is the novelty here?

RESPONSE:

Entropy is not equivalent to “overall changes in expression profile”. Entropy is a well-defined measure that quantifies the total amount of disorder in a transcriptome. In an attempt to make this explicitly clear we have:

- 1) Moved the explicit definitions (Equations 1 and 2) of entropy from the supplementary material to the main text in order to avoid conflation of this concept with more vague definitions.
- 2) Added new figures (Figure 3A and 3B) to better demonstrate the difference between entropy and simpler observations such as large magnitude changes.
- 3) Made multiple changes in the main text (see response to Comment 2 above) to clarify what entropy is and what it isn't.

Moreover, computing transcriptomic entropy by taking into account covariances among gene expression is an approach we present here for the first time. While the definition of entropy for a multivariate gaussian distribution has been established in statistical research fields (5–9), the implementation of entropy in biology, and its application to quantify transcriptional disorder, partially through a loss of transcriptional dependencies, is novel.

Reviewer 1 – Comment 7:

In the analysis shown in Fig. 4, is “model 3” best to predict fitness change? The prediction accuracy of model 3 is almost the same as model 1. If the conclusion is that model 3 is superior to other models, the authors should verify it by appropriate statistical analysis. If it is the case, what is the conclusion by comparing these models 1 to 3?

RESPONSE:

Previously, we had presented 3 variants of the temporal entropy model. Out of these 3, what we had called “model 3” makes the most biologically realistic assumption: that there is a sparse network of regulatory interactions among genes, which affects gene expression patterns over time, and in turn affects overall entropy. We now present the previous “model 3” as the main model (revised Figure 3), and models 1 and 2 as variants of this main model (Supplemental Figure 6).

With the newly added antibiotic datasets, we are now better able to evaluate the performance of temporal entropy on a completely unseen test set. We again observe that the main model where regularization parameter $\rho = 1.5$ (previously model 3), and the variant where $\rho = \infty$ (previously model 1) perform similarly. We use this result to highlight that the entropy-based approach is not as dependent on model parameters (e.g. regularization strength, ρ) as gene-panels. We maintain that the main model (previous model 3) should be favored, as it makes more biologically realistic assumptions. This is discussed in lines 291-298 and 335-339

Reviewer 1 – Comment 8:

In the analysis shown in Fig. 7, the authors should present how the accuracy changes by changing features used for SVM analysis. For example, does SVM using only KLD or entropy (or both) as the features result in lower accuracy? What happens when all the features are used? In addition, what happens when training/test sets are changed for the feature selection? The same feature sets are always selected independently of training/test sets? In this paper, the authors tend to present only one representative result, but results by changing parameters and conditions should also be presented with appropriate statistical analysis to facilitate a better understanding of the generality and robustness of the results.

RESPONSE:

In the previous Figure 7 and the corresponding section in the text, we had tested whether using additional features (such as those obtained from Tn-Seq data, KLD, gene functional category information) would improve predictive performance. We had used a lasso-regularized logistic regression (with a pre-set value for regularization strength) for feature selection, and used the selected features to train an SVM. Taking into account the comments of all reviewers, we revised this analysis, to select features automatically, demonstrate model performance with fewer or more features, and compared multiple models' performance against SVM (this comparison can be found in our response to point 1 from reviewer # 2).

We have now omitted the section on fitness classification using the complex-feature classifier, as it was providing no significant improvement on the entropy-based approach, required more data for predictions, and distracting from the main point of this manuscript, which is the entropy-based predictions.

Regardless, we have performed a round of feature selection to determine whether having more or fewer features would affect performance. In the plot below, the crossvalidation error is shown against the regularization parameter (λ), similar to the feature selection step for the MOA and Fitness-predictor gene-panels. The numbers above the plot indicate the degrees of freedom, or the total number of features corresponding to the different values of λ . The automatic selection of λ in this case resulted in 5 features (including an intercept). With added features, both the mean error on previously unseen data, and the variability in performance (depending on input data) are increased. This indicates that more features do not improve predictive performance, and in fact result in higher error. This is indicative of high variance, or overfitting.

We reiterate that the main advantage of entropy-based models is that they avoid feature selection, since they rely on a single, intuitive feature. Therefore, the fact that in all models that involve feature selection (gene panels for MOA and fitness, and the complex feature classifier) are (to varying degrees) sensitive to the training data makes our case stronger.

Reviewer 1 – Comment 9:

When we consider possible applications of this entropy-based method, the advantage remains unclear. Even when the fitness can be predicted by this method, certainly the transcriptome analysis is expensive compared with direct observation of the fitness (e.g., MIC for antibiotics). If such direct measurement is difficult for some reason, measuring a small number of panel-genes is much easier. Of course, for each microorganism species, seeking appropriate panel-genes is necessary, but it is possible. After finding such panel genes, the experimental cost can be significantly smaller than the entropy-based method proposed in this study. The authors should discuss in what situation this method can be best for analyzing bacterial phenotype.

RESPONSE:

The reviewer touches upon the main motivation behind our entropy-based approach with “Of course, for each microorganism species, seeking appropriate panel-genes is necessary”. We argue that this imposes a very large cost on the gene-panel approach, before it can be applied in clinical diagnostics. For each new antibiotic, and each species, researchers need to collect large amounts of transcriptomic data in order to first determine which genes are relevant for fitness prediction. Moreover, we show that the gene-based approaches are not robust to input data and model parameters, and previously, it has been shown that the isolates used in model training can indeed reduce the applicability of a gene-panel, even within the same species, and for the same stress(10). Our main contribution here is an approach that does not have to be trained on a new species, or a new antimicrobial condition. We have now addressed this point in lines 473-479 in the discussion. Moreover, in lines 496-501 we discuss potential advances in technology that would lower the cost and time required for transcriptome profiling.

Reviewer 1 – Minor comments:

1. In Fig. 1C, I guess the order of vertical labels is incorrect.

RESPONSE:

The labels have been corrected in Figure 2B (corresponding to the old Figure 1C)

2. In page 5, line 129, the gene code of clpP should be presented (sp_0338); otherwise, the readers cannot identify it in Fig. 1D.

RESPONSE:

We have re-written this section, and it no longer references this gene. We have added the locus tag for any genes mentioned in the new text.

3. The authors explain why two strains T4 and T19 are used in this study. What are the differences in the phenotype of these strains? What results are expected by the authors? Is it for checking reproducibility?

RESPONSE:

Both T4 and T19 (Taiwan-19F) originate from clinical isolates, they have approximately 1500 genes in common and each has about 250 unique genes. The main phenotypic difference between T4 and 19F is antibiotic susceptibility: while T4 is sensitive to most antibiotics, Taiwan-19F possess resistance to the penicillin, and a reduced susceptibility to tetracycline and cefepime. Other than this, the two strains (in addition to D39) were selected to avoid any strain-specific bias in the results. Strain specifics are described in the Method section Line 200.

4. The detailed explanation of the D39 strain is helpful in understanding why this strain is used in this study.

5. The vertical axis in Fig. 2A i should be log-scale. The most important information from this growth curve is whether the adapted strains are in an exponential growth phase at the sampling time point (90 min.),

RESPONSE:

The sampling time ranges from 10min-240min depending on the experiment, which is mostly before the exponential phase. The purpose of the growth curves in Figure 1A (previously 2A) and Supplemental Figure 1 is to show the drastic differences between WT and AD strains, and to demonstrate the complete lack of growth in the stress-sensitive strains. We therefore maintained the growth curve representation on a linear scale.

6. The detailed explanation and definition of KLD should be added in Materials and Methods.

RESPONSE:

We have removed the discussion of KLD from the manuscript (see our response to major point 3 above)

Reviewer #2 (Remarks to the Author):

van Opijnen and colleagues conducted a systematic study to determine whether there exists a universal quantifiable feature in bacterial response pattern that is predictive of fitness, irrespective of stress type and species. They developed, optimized and examined a set of predictive models based on experimental evolution experiments and found that the transcriptome entropy, a measure of systems perturbation in face of external stress is predictive of fitness under nutrient and antimicrobial stresses. They also presented a gene-panel based modelling to predict the mechanism of action of antimicrobials with a high accuracy.

The study is timely and takes an innovative approach, in terms of both experimental design and analytical framework. Despite a large literature on diagnostic tests, in particular in the context of genetics of AMR, there has not been much of effort to examine dynamics of transcriptome and this study provides considerable insights into the issue. However, I have significant concerns about the implementation of Machine Learning (ML) methods and justifications of test design and therefore cannot recommend the manuscript for publication in its current format.

A major comment/concern is concerned with details of the implementation machine learning methods throughout the paper. In spite of the importance of the methods in obtaining the final conclusions and results, the predictive methods are not adequately detailed. As a result, it is impossible to assess whether results are reproducible and robust. In the methods section, only a few lines are dedicated to describe the ML methods. In particular the methods section needs to include:

Reviewer 2 – Comment 1:

Method justification: the authors have used SVM without justifying the choice of the method. In two other analyses, logistic regression was used. I wonder if the response feature is sufficiently separated for an SVM to work properly and also whether SVM outperforms simpler logistic regression.

RESPONSE:

We have removed the section regarding the SVM used on complex features (please see our response to Reviewer 1 - Comment 8 above). We did however, repeat the analysis in this section, using automatic feature selection, to address the reviewers' comments. Upon unbiased feature selection, we also performed a comparison of several models (logistic regression (LR), K-nearest neighbor (KNN), decision tree (DT), random forest (RF), and SVM). The performance of each of these models on the training and previously unseen test set is summarized in the table below. Based on AUC of ROC and PR curves, SVM outperforms other models, although based on balanced accuracy and the F1-metric, a decision tree does best.

Model	Group	Cohen's Kappa	AUROC	AUPRC	Sensitivity	Specificity	PPV	NPV	Accuracy	Balanced Accuracy	F1
CFC-DT	Training	0.619784	0.892301	0.937189	0.860927	0.7625	0.872483	0.743902	0.82684	0.811714	0.866667
CFC-DT	Test	0.526316	0.800781	0.565979	1	0.625	0.571429	1	0.75	0.8125	0.727273
CFC-KNN	Training	0.483366	0.865604	0.923049	0.966887	0.4625	0.772487	0.880952	0.792208	0.714694	0.858824
CFC-KNN	Test	0.285714	0.742188	0.46465	1	0.375	0.444444	1	0.583333	0.6875	0.615385
CFC-LR	Training	0.576904	0.838659	0.880393	0.94702	0.5875	0.8125	0.854545	0.822511	0.76726	0.874618

CFC-LR	Test	0.307692	0.773438	0.641492	0.2	1	1	0.666667	0.75	0.6	0.333333
CFC-RF	Training	1	1	1	1	1	1	1	1	1	1
CFC-RF	Test	0.461538	0.882813	0.760001	1	0.5625	0.533333	1	0.708333	0.78125	0.695652
CFC-SVM	Training	0.477785	0.797185	0.864467	0.953642	0.475	0.774194	0.844444	0.787879	0.714321	0.854599
CFC-SVM	Test	0.341463	0.898438	0.856654	1	0.4375	0.470588	1	0.625	0.71875	0.64

Reviewer 2 – Comment 2:

There is no mention of validation dataset throughout the paper and as a result it is unclear whether the model is tested on a completely unseen test dataset.

RESPONSE:

For each prediction approach, we have outlined which experiments were considered in the training set, and which in a previously unseen test set in Supplemental Table 1. Moreover, throughout the manuscript, we had presented (and still present) models split into these training and an unseen test set (examples in the previous Figures 1E-F, 2B-C, 3C-G, 4B-D, Supplemental Figure 5, 6 and 7E). Moreover, Figure 5 (in both the current and the previous version) and the corresponding section describe a second test set that includes completely new set of species (which we had called “validation set”). In each instance, our models are trained only on the training set, and the test set remains unseen during feature selection and parameter tuning. We hope this clarifies any confusion that must have arisen especially because the reviewer also comments on the test set themselves, in Comment 4, and Supplemental Table 1 later.

The only set of results we had previously presented that were not tested on an independent test set were the temporal entropy models 1-3 in previous Figure 4, since we had initially used a subset of our experiments that had more than 4 timepoints. We have since amended this section by including all available experiments in this section, and reporting training and test set performance separately (Figure 3).

Reviewer 2 – Comment 3:

The description of cross-validation to prove that the markers are robust and overfitting has been avoided is missing. Authors need to provide the error rate for test and training data sets. Also, the model ideally should be tested on different held-out datasets to show that feature importance values are robust.

RESPONSE:

For the gene-panel approaches (for prediction of fitness and MOA independently), we now include crossvalidation analysis (new Figure 1B, Supplemental Figure 4B), use this analysis as a basis for selection of features (and number of features) in an automatic and unbiased way, include how feature importance changes with regularization strength and input data (new Figures 1D, E, Supplemental Figures 3B,C,F,G,J,K, Supplemental Figure 4B,C). Our response to Reviewer 1 - Comments 1 and 5 includes detailed explanations of the changes we have made, and where they are found in the manuscript.

Reviewer 2 – Comment 4:

Hyperparameter tuning and details of optimized model are absent. Also given the limited number of samples in the test dataset for predicting MOA, as indicated in Figure 1, I wonder if authors have

introduced any regularization in the SVM methods. This can include L2 and C regularization and using kernel methods.

RESPONSE:

We had presented 3 types of models: logistic-regression (i.e. gene-panels), entropy-based models, and the SVM in the complex feature classification section (corresponding to the previous Figure 7). For the logistic regression models (gene-panels), we now present a justification for selecting the regularization parameter by using crossvalidation error. We use well-established principles for the tuning of this parameter, by selecting the simplest model (i.e. model with fewest features) that achieves comparable performance to the best possible model (1,2). In the revised manuscript, we only include 2 types of models (logistic regression and entropy-based), and tune the regularization parameter when applicable (Figure 1B, Supplemental Figure 3A, E, I, Supplemental Figure 4A, Figure 3C) using this established principle.

As mentioned, although we have removed the complex-feature classifier (previously SVM) from the manuscript, we did try multiple models, and hyperparameter tuning for each model. The parameter grids we scanned, and the resulting best model (based on crossvalidation) can still be found in the jupyter notebook: <https://github.com/dsurujon/FitnessPrediction/blob/master/CFC%20models.ipynb>

Reviewer 2 – Comment 5:

To report the performance of a diagnostic tool, accuracy is not informative and other prediction metrics, e.g. AUC, specificity, recall and precision, should be used.

RESPONSE:

While accuracy is indeed an important and informative metric, we report AUC (on ROC and PR), specificity, recall, precision, F1 and kappa statistics for all prediction models in Supplementary Table 8.

Reviewer 2 – Comment 6:

The choice of six genes in Line 333 is not justified. Also feature importance analysis requires an examination of co-linearity between features, which is not provided. It is unclear why authors decided to reduce predictor features without assessing the extent of overfitting when all features are used.

RESPONSE:

We have included more detail on feature selection for each model we retain in the revised manuscript. Please see our answers to Reviewer 1 - Comments 1 and 5, and Reviewer 2 - Comments 3 and 4.

Reviewer 2 – Comment 7:

The choice of 8 genes (Line 122), later 10 genes (Line 193) in the gene panel seem arbitrary and have been weakly justified. I expect the authors to conduct a sensitivity analysis to demonstrate how inclusion of more gene will affect the predictive power and whether mining the input data into few genes has not resulted in a significant information loss. By doing so, authors can show that their threshold/cut-off choice for markers numbers is objective.

RESPONSE:

We have included a feature importance and prevalence analysis for the gene panel models. Our new model selection scheme also selects the regularization strength (i.e. number of features)

objectively, based on crossvalidation error within the training set. See our response to Reviewer 1-Comments 1 and 5 and Reviewer 2 - Comments 3 and 4.

Reviewer 2 – Comment 8:

Given that authors have studied only a few representative strains, I think the conclusion on the applicability of prediction power of entropy to other conditions/species is a bit far-fetched. Importantly, the effect of genetic background is not discussed in the limitations. Also, from supplemental table S1, it seems the method has been examined only on few strains from other species and therefore the results are not generalizable to the population level.

RESPONSE:

We respectfully disagree with the reviewer on this point. We had already demonstrated that entropy-based classification works on 5 new species in Figure 5, which we had selected specifically not to limit ourselves to a small number of lab strains. These strains include clinical isolates from CDC, and both Gram negative and Gram positive species, representing a vastly different set of genetic backgrounds. In order to further support the argument that entropy is generalizable, we have now added the analysis of a published dataset(11) coming from 3 species, 36 strains in total, exposed to 3 antibiotics (2 of them not previously included in our dataset). The revised Figure 5E shows that entropy successfully separates resistant strains from susceptible ones in this new dataset as well.

Additionally, entropy models are based on a first-principle approach and thus do not require a large training set – the only parameter requiring training is the entropy threshold. By demonstrating that the entropy threshold trained from *S. pneumoniae* data (and only on the training portion of our dataset) allowed for fitness predictions on 5 species at an 100% accuracy, we believe it is fair to claim that our entropy model is generalizable.

Reviewer 2 – Comment 9:

Furthermore, in the limitation section, authors should mention that the entropy metric has been tried in a controlled experimental evolution condition in the lab, which does not necessarily represent natural and clinical conditions.

RESPONSE:

As mentioned in the response to the previous point, we have included several clinical isolates in our existing dataset, and in the new dataset from (11). We have also included a discussion of how we envision entropy to be applicable in the clinic in lines 496-501.

Reviewer 2 – Comment 10:

From a clinical point of view, a diagnostic test should not only have a high accuracy but provide a mechanistic understanding of the underlying features. The link between entropy and stress does not seem to have the latter or at least this is not adequately mentioned in discussion.

RESPONSE:

We acknowledge and appreciate the value in gaining biological insights, even though this is rarely important for a diagnostics test. In terms of clinical applicability, we argue that diagnostic approaches that rely on specific mechanisms will eventually be limited to specific strains/species or conditions, and these approaches need to be re-implemented and re-trained for each new setting. Moreover, our results indicate that there may not be a general stress response signature that could be extracted from specific genes in transcriptome data for two main reasons: 1) it is possible to

select different sets of genes that have similar predictive performance in a gene-panel on previously unseen data, and 2) in the selected gene-panels, there is little functional enrichment. We have now included enrichment analyses that show there is no functional enrichment (which may lead to new biological hypotheses about the action of the stress/antibiotic) in more general (i.e. condition-agnostic) gene-panels, but there is enrichment only when a gene-panel is specific to a single drug, in a single species (lines 199-210). This highlights the trade-off between a model that captures descriptive, mechanistic understanding on antibiotic action, and a model that is universal. However, we now include a discussion on how the co-expression networks that are generated (as an intermediate step) while calculating entropy may have valuable information in lines 479-484. Additionally, our entropy based-models highlight that increasing stress leads to an increasing loss in transcriptional dependencies between genes, which to our knowledge is an unknown biological insight that merits further investigation.

Other comments:

Lines 194-202 include descriptive and inconclusive results. The section looks a bit incomplete and can be either shortened or left out.

RESPONSE:

This section is no longer in the revised manuscript.

Line 209 ESKAPE should be defined.

RESPONSE:

We no longer use the abbreviation ESKAPE. We refer to each pathogen by their species name.

Line 456 Support Vector Machine should be capitalized.

RESPONSE:

We no longer include the Support Vector Machine predictions.

Lines 163-173 can be moved to Methods.

RESPONSE:

We maintain that this information is critical for explaining the experimental setup, thus we have kept it in the main text (now lines 155-164)

Line 132-140 The functional enrichment analysis is unconvincing and sounds like cherry-picking. It is unclear whether or not the occurrence of the genes merely reflects the underlying distribution of gene functions.

RESPONSE:

We have now included an unbiased enrichment analysis on all our gene-panels (Supplemental Figure 2E, Supplemental Figure 3D, H, L, Supplemental figure 4E), as well as a set of 9 published gene-panels(11) Supplemental File 2). We omit the discussion of individual genes from the main text.

Lines 231-250 can be significantly shortened.

RESPONSE:

We have edited this section substantially, as it highlights the novelty in our approach, and is therefore a critical section of the manuscript. We now present the previous Model 3 as the main entropy approach and all other entropy-based approaches as its variants, in order to keep this section clear.

Lines 82-85 sound unclear and should be rephrased/elaborated, given the importance of the message.

RESPONSE:

We have re-written this section to only reference the relevant results in the published studies (see lines 84-90).

Reviewer #3 (Remarks to the Author):

In this work, the authors aim to construct models to classify bacterial strains as sensitive or resistant to antibiotics and predict their MIC (Minimal Inhibitory Concentration) using global statics of their transcription response to antibiotics. First, the authors demonstrate the limited predictive capabilities of specific-gene models. Second, they demonstrate that a variety of environmental stresses induce widespread transcriptional changes in WT (stressed) versus pre-adapted (unstressed) populations, and thus, can be used to classify populations and predict MIC. In particular, they introduce the notion of predictions based on “entropy” - the degree to which transcriptional changes occur due to antibiotic treatment. Third, they incorporated additional data from Tn-Seq and showed it can be used to improve predictions.

Major comments:

Reviewer 3 – Comment 1:

Overall, I have difficulty verbalizing the main contribution and novelty of this paper. I am worried (but not entirely sure) that the paper takes simple notions and wrap them in vague sexy terms. If I try to peel away the big words of “entropy” etc, the primary biological statement that emerges is rather simple and expected: the antibiotics cause more stress to the antibiotic sensitive cells than to the antibiotic resistant cells, leading to vast differences in their transcriptional response. This observation does not sound particularly surprising or novel to me. The claim that the authors discovered a new general signature of bacterial stress response therefore seems overblown. For a number of bacterial species, it is known that antibiotic exposure leads to genome-wide changes in gene expression due to both generalized and specific stress responses (e.g., Gottesman, 2019). I view “entropy” as a way to quantify this stress response, but the underlying phenomenon is not new. I want to be careful here: I may be missing something profound and if so I would be glad to be corrected.

RESPONSE:

While we use a possibly relatively simple intuition (a more stressed out transcriptome is more disordered), we define entropy in a particular way that is distinct from “vast differences in the transcriptional response”. We now highlight in the revised manuscript how entropy, which quantifies disorder by taking into account dependencies among genes, is different from simply large transcriptional changes. Specifically, in lines 104-115, 265-281, 444-449, and Figure 3B, we specify the definition of entropy, and highlight how it is different from “large changes in the transcriptome”.

We acknowledge that there are similarities between entropy and less specific findings such as what has been observed before, as the reviewer notes. We now include lines 88-93, where we highlight the large transcriptomic differences that have been observed, in order to further justify our choice of this phenomenon as something to focus on. This work is the first to take this intuitive observation, carefully defines a metric (entropy) that is nuanced, and corrects for regulatory dependencies, and produces a predictive tool using this metric.

We also stress that our definition of entropy is very much the same as the statistical definition of the entropy of a multivariate (or univariate) gaussian distribution. As such entropy has been used multiple publications from the field of statistics prior to this study (5–9). Entropy is thereby a well-established feature of such distributions, similar to mean and variance. For this reason, we respectfully disagree with the reviewer that the entropy is a “vague sexy term” that is used inappropriately. However, we acknowledge that this may have arisen from us not explaining entropy clearly in the main manuscript. We have made sure to rectify this by making multiple changes throughout the manuscript and adding additional figures to make sure it is clear what entropy means.

The generalized and specific responses discussed in (12) center around the sigma factor *rpoS*. While such responses have been characterized for species such as *E. coli*, in others such as *S. pneumoniae*, a homolog of *rpoS* has not been identified. Importantly, the activation of this ‘general response’ depends heavily on the stress factor, and whether there is any overlap in the downstream transcriptional changes (triggered by different stressors) has not been shown. Our data suggests that there is no common stress-signature that depends on specific genes. Instead we find that entropy can be used to quantify global transcriptional disorder by taking into account dependencies among genes. We have included lines 73-78 and 454-464 to discuss ‘the general response’ and how it relates to this work.

Reviewer 3 – Comment 2:

The primary application – transcriptome-based diagnostics – doesn’t seem practical, at least not in the near future. And if it does turn practical, I am most certain that an approach based on ML model trained on the gene-specific transcriptional response (rather than global statistics) of several species to different antibiotics will do much better. Even if it does show superiority, I would also appreciate a more detailed discussion of how the authors envision their method being used in the clinic, since I have a hard time envisioning its practical utility. Specifically, the proposed approach requires measuring a full transcriptome, which seems more expensive and more time-consuming than simply measuring the MIC.

RESPONSE:

While measuring the full transcriptome can be costly and time consuming with the current technology, we maintain that it would be faster to perform for slow-growing species such as *M. tuberculosis* compared to culture-based MIC testing (which could take weeks). Moreover, the data collection for the reviewer’s proposed ML approach (including training data from multiple species and conditions) would be arguably more expensive, as it requires full transcriptome data from many species, and many antibiotic conditions. We elaborate on this point in lines 473-479. Furthermore, for such an approach, it would not be clear whether the sampling in the training data has been sufficiently diverse (in terms of species and conditions), unless many more transcriptomic data are collected as a test set to validate this approach. While these types of methods do have the potential of getting better with additional training data, the amount of training data required may be prohibitively large. What we propose is a method that has the capacity to generalize with a limited set of training data (as we show in the manuscript).

We have implemented a similar approach to what the reviewer proposes in our condition-agnostic gene-panel based fitness predictions (although we acknowledge that this included multiple conditions, but not multiple species). We have shown that this approach is not robust to input data, and model parameters (Figure 1). Moreover, training an ML model that is applicable on multiple species has an additional limitation: the lack of homology across species. The more species are considered, the number of genes that are shared across these species will decrease, which might make it more difficult to extract valuable information from the few genes that are in common.

In addition, we have added a discussion on emerging technologies that would make transcriptome profiling faster and cheaper in lines 496-501. Lastly, making predictions on what will and what will not be useful in the future is a tricky business and rarely holds. Our entropy-based approach shows that it is possible to make fitness and/or MIC predictions in a very different manner than what is done so far. Thereby this adds diversity to our way of thinking, it opens up new possibilities for the future, moreover it has given us the biological insight that increasing stress comes with increasing loss of dependencies among genes, which is a really interesting insight to explore in follow-up research.

Minor comments:

1. The term “entropy” is more confusing than helpful to me. As I understand it, “entropy” simply quantifies the variance in the $\log_2(\text{gene expression} \pm \text{drug})$ distribution. However, it was difficult to parse this from the text. Furthermore, given the variety of meanings that “entropy” can take, the term is easily misinterpreted, depending on the reader’s background. Perhaps the term “stress-induced transcriptional change” would be more transparent.

RESPONSE:

For both the single timepoint and the temporal models, we use the statistical definitions of entropy (defined for a univariate and multivariate gaussian distribution, respectively). Therefore, we maintain the nomenclature is appropriate (while also being succinct). We have also included several references that use the same well-established definitions for entropy in the field of statistics (5–9).

Importantly, entropy on a temporal transcriptomic dataset is defined as a metric that accounts for condition-specific dependencies among genes, which is more sophisticated than the variance of a single-timepoint. We now present the temporal entropy (Equation 1 in the main text) as the central idea, and the single-timepoint variant as something that could be used when the data are limited (i.e. only a single timepoint is available).

2. Fig. 1C: One of the axes is flipped.

RESPONSE:

This has now been fixed (new Figure 2B)

3. In multiple cases, the same data is shown via two different representations, but to support the same overall message. For example, Fig. 2B and 2C are redundant. Removing these panels will help to streamline the paper.

RESPONSE:

We have made substantial changes to all main and supplemental figures, and removed redundant information from the main figures.

References

1. Friedman J, Hastie T, Tibshirani R. Regularization Paths for Generalized Linear Models via Coordinate Descent. *J Stat Softw.* 2010;33(1):1–22.
2. Krstajic D, Buturovic LJ, Leahy DE, Thomas S. Cross-validation pitfalls when selecting and assessing regression and classification models. *Journal of Cheminformatics.* 2014 Mar 29;6(1):10.
3. Barenholz U, Keren L, Segal E, Milo R. A Minimalistic Resource Allocation Model to Explain Ubiquitous Increase in Protein Expression with Growth Rate. *PLOS ONE.* 2016 Apr 13;11(4):e0153344.
4. Kaneko K, Furusawa C, Yomo T. Universal Relationship in Gene-Expression Changes for Cells in Steady-Growth State. *Phys Rev X.* 2015 Feb 11;5(1):011014.
5. Ahmed NA, Gokhale DV. Entropy expressions and their estimators for multivariate distributions. *IEEE Transactions on Information Theory.* 1989 May;35(3):688–92.
6. Misra N, Singh H, Demchuk E. Estimation of the entropy of a multivariate normal distribution. *Journal of Multivariate Analysis.* 2005 Feb 1;92(2):324–42.
7. Cai TT, Liang T, Zhou HH. Law of log determinant of sample covariance matrix and optimal estimation of differential entropy for high-dimensional Gaussian distributions. *Journal of Multivariate Analysis.* 2015 May 1;137:161–72.
8. Srivastava S, Gupta MR. Bayesian estimation of the entropy of the multivariate Gaussian. In: 2008 IEEE International Symposium on Information Theory. 2008. p. 1103–7.
9. Lazo AV, Rathie P. On the entropy of continuous probability distributions (Corresp.). *IEEE Transactions on Information Theory.* 1978 Jan;24(1):120–2.
10. Wadsworth CB, Sater MRA, Bhattacharyya RP, Grad YH. Impact of Species Diversity on the Design of RNA-Based Diagnostics for Antibiotic Resistance in *Neisseria gonorrhoeae*. *Antimicrobial Agents and Chemotherapy* [Internet]. 2019 Aug 1 [cited 2019 Dec 23];63(8). Available from: <https://aac.asm.org/content/63/8/e00549-19>
11. Bhattacharyya RP, Bandyopadhyay N, Ma P, Son SS, Liu J, He LL, et al. Simultaneous detection of genotype and phenotype enables rapid and accurate antibiotic susceptibility determination. *Nature Medicine.* 2019 Nov 25;1–7.
12. Gottesman S. Trouble is coming: Signaling pathways that regulate general stress responses in bacteria. *J Biol Chem.* 2019 Jun 13;jbc.REV119.005593.

Reviewers' Comments:

Reviewer #1:

Remarks to the Author:

I found that the revised manuscript is much improved. Notably, the authors thoroughly revised the explanation of the concept of "entropy," and now the difference between the entropy and the differential expression (DE) is clear. However, still, this paper is difficult to understand, and the novelty and applicability of this method remain questionable.

Comments:

i) Based on several figures (e.g., Figure 3B and Supplemental Figure 5) and the explanations, I understood the concept of the entropy in the time-series analysis. I agreed that it can be a novel approach to extract a kind of complexity in the expression dynamics. In contrast, for the single timepoint data, the entropy is simply the variance of the DE distribution. Here, as the authors wrote in reply to the reviewer #3, the central idea and novelty in this study lay the temporal entropy. In contrast, the single timepoint entropy is not essential in this study. If it is the case, the authors should remove all results using single timepoint data to make the concept of this study easy to understand.

ii) The authors explained the difference between DE and the entropy in Fig. 3B, in which the complex expression dynamics as "scenario 3" will have larger entropy than the monotonical expression changes with a similar timescale, as "scenario 2". I agree that it would be quite interesting if such a difference between scenarios 2 and 3 can be observed in the time-series expression data depending on the fitness—however, the time-series data of Fig. 3A (left figure) looks like scenario 2, i.e., the expression levels monotonically change with the similar timescale, even though the fitness is low. Although there are a small number of exceptions showing non-monotonical expression changes, I wonder whether only these genes contribute to the high temporal entropy in the low fitness data, or more complex dynamics exists in other time-series data. In addition to statistics of expression profiles, such as the entropy, the authors should present examples of "chaotic gene expression changes" more explicitly and discuss what kind of gene functions are involved in the chaotic expression dynamics.

iii) It would be interesting if the complex expression dynamics, as scenario 3 in Fig. 3B, is found in the time-series expression data of low fitness cases. However, in order to accept the complexity of expression dynamics, it is necessary to show that it is not generated by experimental error. To observe complex expression dynamics, as in scenario 3, using mRNA samples extracted from millions of cells, the expression profiles of a certain fraction of cells need to change synchronously. Otherwise, such complex expression dynamics should disappear due to averaging out. If the expression changes become more stochastic in low fitness state, as the authors wrote in the fifth sentence in the discussion section, such dynamics seem difficult to identify by transcriptome data, which corresponds to the average expression levels of millions of cells. The authors need to carefully show how the complex dynamics as scenario 3 can be quantified and how it can be discriminated from experimental noise using their experimental data. Checking the reproducibility of complex expression dynamics and using appropriate statistical analysis are necessary.

iv) As mentioned above, to obtain the temporal entropy, careful experiments to quantify time-series expression data (e.g., with the low noise level and high reproducibility) and appropriate way to discriminate the signal and noise are necessary. This fact means that to use the temporal entropy requires quite high experimental cost, which limits the applicability of the proposed method. I wonder what kind of targets the temporal entropy with high cost should be used. In fact, according to Supplemental Table 8, the prediction accuracy by single timepoint entropy, corresponding to the variance of DE distribution, seems similar to that of the temporal entropy. This result suggested that there is no room to use the temporal entropy, which requires careful and costly experiments.

Reviewer #2:

Remarks to the Author:

The authors have significantly improved the manuscript. My main concern was about the implementation of predictive models, which have now been addressed. The approach is now justified. Also, there are more validations and justifications with other datasets. I enjoyed reading the new version of the paper.

Reviewer #3:

Remarks to the Author:

The authors have adequately answered all of my comments and suggestions. This manuscript is much improved especially in the way it contrasts the current approach with previous literature. The concept of entropy is now also much better explained. I still cannot see how this approach will become practical and I am also quite certain that if RNA-seq based diagnostics does appear it will be the more straight forward machine learning models that act on the entire transcriptome (rather than the entropy) that will prevail. Yet, I think this is scientifically an interesting line of thought and will trigger discussion and further research. I therefore recommend and looking forward to seeing this work on the pages of Nature Communications.

We thank the reviewers for their consideration and positive feedback. While reviewers 2 and 3 had no further comments and are supportive of publication, we made several changes to the main text, and added a supplemental figure in order to address specific comments by Reviewer 1. Below we specify the changes made, and how they relate to each specific comment.

Reviewer #1 (Remarks to the Author):

I found that the revised manuscript is much improved. Notably, the authors thoroughly revised the explanation of the concept of "entropy," and now the difference between the entropy and the differential expression (DE) is clear. However, still, this paper is difficult to understand, and the novelty and applicability of this method remain questionable.

Comments:

Reviewer 1 – Comment 1:

Based on several figures (e.g., Figure 3B and Supplemental Figure 5) and the explanations, I understood the concept of the entropy in the time-series analysis. I agreed that it can be a novel approach to extract a kind of complexity in the expression dynamics. In contrast, for the single timepoint data, the entropy is simply the variance of the DE distribution. Here, as the authors wrote in reply to the reviewer #3, the central idea and novelty in this study lay the temporal entropy. In contrast, the single timepoint entropy is not essential in this study. If it is the case, the authors should remove all results using single timepoint data to make the concept of this study easy to understand.

RESPONSE

The novelty in this manuscript lies in the development of transcriptional entropy as a universal measurement of stress in bacteria, whose level is indicative of a bacterium's fitness in an environment. Entropy can be determined from temporal data giving detailed information of the state of the organism at each time point. However, as the reviewer points out in their Comment #4, multiple-timepoint experiments might have prohibitive cost associated with them. Moreover, they can be complex, labor intensive and time consuming. With the goal to reduce the overall complexity of temporal experiments, we found that temporal entropy can be reduced in its complexity to a single timepoint model, without compromising predictive performance. The novelty in this manuscript is thus extended to the beauty that lies in the simplicity of the single-time point approach. To highlight this, we have included the following statement:

P13L375: This simpler definition of entropy enables the approach to be applied even in settings where temporal transcriptional information cannot be obtained.

Furthermore, with the single-timepoint entropy, we show that predictions can be extended from a binary fitness predictor to a predictor of minimum inhibitory concentration (MIC). This result emphasizes the flexibility of our approach, and the added information given by the single timepoint model. To highlight this, we made a small change to the sentence on P15L447 and added the following statement to the discussion:

P17L487-490: This study demonstrates how entropy-based predictive models can be implemented in several ways, by using different amounts of data, resulting in different types of predictions. Even using a single timepoint, it is possible to predict both fitness as a binary outcome, as well as the MIC of an antibiotic (Figure 5D), highlighting entropy as a flexible framework that can be adapted to different settings.

Reviewer 1 – Comment 2:

The authors explained the difference between DE and the entropy in Fig. 3B, in which the complex expression dynamics as “scenario 3” will have larger entropy than the monotonical expression changes with a similar timescale, as “scenario 2”. I agree that it would be quite interesting if such a difference between scenarios 2 and 3 can be observed in the time-series expression data depending on the fitness—however, the time-series data of Fig. 3A (left figure) looks like scenario 2, i.e., the expression levels monotonically change with the similar timescale, even though the fitness is low. Although there are a small number of exceptions showing non-monotonical expression changes, I wonder whether only these genes contribute to the high temporal entropy in the low fitness data, or more complex dynamics exists in other time-series data. In addition to statistics of expression profiles, such as the entropy, the authors should present examples of "chaotic gene expression changes" more explicitly and discuss what kind of gene functions are involved in the chaotic expression dynamics.

RESPONSE:

In Figure 3A, there are many overlapping lines, and visually it is impossible to pick up on differential expression patterns that are not the largest in magnitude and determine whether the example fits with any of the scenarios. Importantly, the scenarios function as examples to illustrate how entropy can be high or low, any biological system will have a mixture of many scenarios, just like the example in Figure 3A will consist of many scenarios. To not be dependent on an experimenter’s interpretation of what they may think is the most likely scenario is a key reason why we use objective models for interpretations, which here results in the detailed development of entropy. To make sure no misunderstanding will occur and to emphasize this for the reviewer and the reader we specifically determined whether the vast majority of genes follow a monotonic increase or decrease by performing clustering on the temporal gene expression data for the T4 VNC experiment. Using Fuzzy c-means clustering¹, genes were grouped into clusters where each cluster is composed of genes with matching expression profiles, distinct from genes outside of the cluster.

As the true number of clusters K_{True} is unknown, the number of clusters was estimated using the elbow method². For the range $K = 3, 4, \dots, 30$ clusters, the minimal distance between cluster centroids (D_{min}) was computed (Figure 1). K_{True} is then estimated to be the elbow point in the D_{min} vs K curve. The intuition behind this is as follows. When $K < K_{True}$, increasing K results in a decline in D_{min} . When $K > K_{True}$, increasing K results in the splitting of real clusters, yielding smaller, highly similar clusters. The separation of clusters is minimized once K_{True} is exceeded, and the decrease in D_{min} is less drastic.

Figure 1. Minimal centroid distance (D_{min}) decreases with increased number of clusters (K). There is a clear elbow point at $K = 10$.

For the T4 VNC experiment, the elbow point is at $K = 10$ (Figure 1). Clustering of gene differential expression patterns was done using this value (Figure 2). Out of the 10 clusters, 4 (Clusters 2, 5, 9 and 10) have a monotonic gene expression pattern, and the curvature of clusters 2 and 5 are highly distinct from clusters 9 and 10. The total number of genes in these 4 clusters make up 45% of the entire transcriptome. This demonstrates that the gene expression profiles are not dominated by monotonic increase/decrease patterns, and that there is a mix of vastly different temporal expression profiles.

Figure 2. Standardized differential expression over time for each gene, in each cluster. Each line represents an individual gene, and the color of the line indicates the degree of similarity to the cluster. Yellow indicates lower similarity, blue medium, and magenta indicates high similarity to the cluster.

We would like to stress the importance of differences in expression profiles as the main contributor to entropy. If the entire transcriptome looked like Cluster 1 or 4 (Figure 2), that would still be a low-entropy response, because the expression patterns of all the genes follow a similar trajectory. What matters most is the level of similarity (i.e. high or low covariance) in expression profiles of different genes, not what that profile looks like.

The 10 clusters above also do not associate very strongly with a certain function (Figure 3). This suggests that specific functions do not have a single type of temporal expression pattern enriched. Rather, each function is an approximately even mix of all types of expression patterns. Thus, it is likely that each function has a similar contribution to overall entropy in this experiment.

Figure 3. All clusters of distinct expression patterns are present in each cellular functional category, in similar proportions.

In order to determine whether specific functional groups contributed more to entropy, temporal entropy was computed for each set of genes belonging to a particular function (Figure 3). The entropy values obtained from different functions were highly similar for the same experiment. Since there is little difference between the entropy of individual functions, it is unlikely that genes belonging to a certain function contribute more to entropy.

Figure 3 (Supplemental Figure 6). Each of the 55 experiments is represented by a line. Temporal entropy with $\rho=\infty$ was computed on each of the 5 sets of genes belonging to different cellular functions. The entropy values computed on the 5 non-overlapping gene sets separated by functional tags are similar for each experiment.

We have included Figure 3 as the new Supplemental Figure 6. The point made above is also mentioned in the main text:

P12L337: Moreover, entropy of each cellular function is similar for a given experiment (Supplemental Figure 6), suggesting that transcriptome-wide entropy is not dominated or influenced by a certain set of genes.

Reviewer 1 – Comment 3:

It would be interesting if the complex expression dynamics, as scenario 3 in Fig. 3B, is found in the time-series expression data of low fitness cases. However, in order to accept the complexity of expression dynamics, it is necessary to show that it is not generated by experimental error. To observe complex expression dynamics, as in scenario 3, using mRNA samples extracted from millions of cells, the expression profiles of a certain fraction of cells need to change synchronously. Otherwise, such complex expression dynamics should disappear due to averaging out. If the expression changes become more stochastic in low fitness state, as the authors wrote in the fifth sentence in the discussion section, such dynamics seem difficult to identify by transcriptome data, which corresponds to the average expression levels of millions of cells. The authors need to carefully show how the complex dynamics as scenario 3 can be quantified and how it can be discriminated from experimental noise using their experimental data. Checking the reproducibility of complex expression dynamics and using appropriate statistical analysis are necessary.

RESPONSE:

We would like to stress that all experiments have been executed carefully, with appropriate controls, and according to accepted and widely used standards. We have gone to extreme lengths to make our data easily accessible and available for anyone to explore. All experiments, comparisons and analyses have been performed in extreme detail resulting in the development of entropy that we extensively validated across strains, species and entirely different (and unrelated) datasets. All experiments were tested for their reproducibility with extremely high accuracies. Importantly, these accuracies can even be explored by the reader in our accompanying browser-accessible instance of ShinyOmics. To further highlight the reproducibility of our experiments, we have included a correlation analysis across biological replicates for each timepoint within each experiment. Each experimental timepoint had 3 or 4 replicates, and we computed the Spearman correlation between two replicates, for all possible pairwise comparisons (3 or 7 comparisons respectively). We have added the following statement in the Supplemental Methods section titled “Temporal RNA-Seq sample collection, preparation and analysis” (p. 22) referring to this analysis and highlighting the reproducibility across replicates:

The reproducibility of the transcriptomic data was confirmed by an overall high Spearman correlation across biological replicates ($R > 0.95$). Furthermore, the consistent patterns we observe in DE for the training, test and validation experiments, as well as the similarity of DE from experiments using antibiotics with the same MOA, point to the high quality and reproducibility of the dataset. NB: comparison of experiments can be done using ShinyOmics (<http://bioinformatics.bc.edu/shiny/ABX>).

Reviewer 1 – Comment 4:

As mentioned above, to obtain the temporal entropy, careful experiments to quantify time-series expression data (e.g., with the low noise level and high reproducibility) and appropriate way to discriminate the signal and noise are necessary. This fact means that to use the temporal entropy requires quite high experimental cost, which limits the applicability of the proposed method. I wonder what kind of targets the temporal entropy with high cost should be used. In fact, according to Supplemental Table 8, the prediction accuracy by single timepoint entropy, corresponding to the

variance of DE distribution, seems similar to that of the temporal entropy. This result suggested that there is no room to use the temporal entropy, which requires careful and costly experiments.

RESPONSE:

Indeed, computing temporal entropy requires more experiments, and would increase the cost proportional to the number of time points. However, we have shown that using as few as 2 time points is sufficient to discriminate between high and low fitness (Supplemental Figure 8), and thus to implement temporal entropy. The more time points included would decrease uncertainty in the coexpression - measured as covariance values - across genes, and therefore the temporal entropy. Any experiment and approach requires a researcher to evaluate the funds and resources they have available, leading to an optimization of experiments:funds ratio. We have addressed this as follows:

P19L541-548: In particular, time-course experiments such as those included in this study increase in cost linearly with an increasing number of time points. However, the advances in technology are likely to reduce cost much more drastically than a linear model, as is observed for many sequencing approaches. To implement temporal entropy, it is important to recognize that more timepoints will yield better results. However, even 2 timepoints gives robust results. The most economic approach would clearly be the single-timepoint model, which has comparable performance to the temporal models, with the only disadvantage that it lacks possible insights that could be gleaned from the covariance networks temporal entropy is based on.

We also stress that this approach may lead to new discoveries. Although it is not the scope of this manuscript, the covariance networks produced using the temporal entropy approach are a new way of exploring transcriptional dependencies. We mention this point in lines 517-522. To be specific, we are currently studying whether the hubs on these covariance networks correspond to transcription factors or genes involved in regulation, how much overlap there is in the networks generated in the different antibiotic conditions, and whether the network structure is predictive of transcriptomic state of a bacterium given a specific environmental perturbation. The temporal entropy approach therefore has potential beyond fitness predictions. We added a brief sentence to address this point:

P18L522-524: Thus, it is possible that the networks generated in this work will be applicable in other ways, e.g. in the identification of novel regulators, their targets, or the prediction of transcriptional changes that follow a perturbation.

Reviewer #2 (Remarks to the Author):

The authors have significantly improved the manuscript. My main concern was about the implementation of predictive models, which have now been addressed. The approach is now justified. Also, there are more validations and justifications with other datasets. I enjoyed reading the new version of the paper.

Reviewer #3 (Remarks to the Author):

The authors have adequately answered all of my comments and suggestions. This manuscript is much improved especially in the way it contrasts the current approach with previous literature. The concept of entropy is now also much better explained. I still cannot see how this approach will become practical and I am also quite certain that if RNA-seq based diagnostics does appear it will be the more straight forward machine learning models that act on the entire transcriptome (rather than the entropy) that will prevail. Yet, I think this is scientifically an interesting line of thought and will trigger discussion and further research. I therefore recommend and looking forward to seeing this work on the pages of Nature Communications.

References

1. Kumar, L. & E. Futschik, M. Mfuzz: A software package for soft clustering of microarray data.

Bioinformatics 2, 5–7 (2007).

2. Thorndike, R. L. Who belongs in the family? *Psychometrika* 18, 267–276 (1953).

Reviewers' Comments:

Reviewer #1:

Remarks to the Author:

The authors significantly improved the manuscript. My concerns have now been addressed. I therefore recommend it for the publication in Nature Communications.